# Pseudoknot length modulates the folding, conformational dynamics, and robustness of Xrn1 resistance of flaviviral xrRNAs

Xiaolin Niu [1,5], Ruirui Sun [1,2,5], Zhifeng Chen[1,3], Yirong Yao[1,2], Xiaobing Zuo [4], Chunlai Chen[1,2✉] & Xianyang Fang [1✉]

To understand how RNA dynamics is regulated and connected to its function, we investigate the folding, conformational dynamics and robustness of Xrn1 resistance of a set of flaviviral xrRNAs using SAXS, smFRET and in vitro enzymatic assays. Flaviviral xrRNAs form discrete ring-like 3D structures, in which the length of a conserved long-range pseudoknot (PK2) ranges from 2 bp to 7 bp. We find that xrRNAs' folding, conformational dynamics and Xrn1 resistance are strongly correlated and highly $Mg^{2+}$-dependent, furthermore, the $Mg^{2+}$-dependence is modulated by PK2 length variations. xrRNAs with long PK2 require less $Mg^{2+}$ to stabilize their folding, exhibit reduced conformational dynamics and strong Xrn1 resistance even at low $Mg^{2+}$, and tolerate mutations at key tertiary motifs at high $Mg^{2+}$, which generally are destructive to xrRNAs with short PK2. These results demonstrate an unusual regulatory mechanism of RNA dynamics providing insights into the functions and future biomedical applications of xrRNAs.

[1] Beijing Advanced Innovation Center for Structural Biology, School of Life Sciences, Tsinghua University, Beijing 100084, China. [2] Beijing Frontier Research Center for Biological Structure, School of Life Sciences, Tsinghua University, Beijing 100084, China. [3] State Key Laboratory for the Chemistry and Molecular Engineering of Medicinal Resources, School of Chemistry and Pharmaceutical Sciences, Guangxi Normal University, Guilin 541004, China. [4] X-ray Science Division, Argonne National Laboratory, Lemont, IL 60439, USA. [5] These authors contributed equally: Xiaolin Niu, Ruirui Sun.
✉email: chunlai@mail.tsinghua.edu.cn; fangxy@mail.tsinghua.edu.cn

Many RNAs, including the noncoding RNAs, the untranslated and/or coding regions of viral genomic RNAs, fold into complex and unique 3D structures[1–3]. Besides, RNA molecules are inherently flexible and dynamic, which adaptively acquire distinct 3D structures on their own or in response to a diverse array of cellular conditions[4]. RNA structural dynamics is then linked to its diverse cellular functions[5], such as serving as templates for the synthesis of proteins, RNAs, or DNAs, as catalytic centers for transcription, pre-mRNA splicing, translation, or as regulators of gene expression[6]. Numerous RNA tertiary motifs including coaxial helical stacking, long-range pseudoknots, and kissing loops and their contributions to RNA 3D structure have been revealed[7,8], but how these motifs modulate RNAs' structural dynamics to achieve their diverse functions remains elusive.

Exoribonuclease-resistant RNAs (xrRNAs) are discrete RNA elements that block the processive exoribonucleolytic degradation of RNA[9]. xrRNAs were originally identified at the beginning of the 3′ untranslated region (UTR) of genomic RNAs of several mosquito-borne flaviviruses (MBFVs), including Dengue virus (DENV), Zika virus (ZIKV), and West Nile virus (WNV), which enable the generation of noncoding subgenomic flaviviral RNAs (sfRNAs) in infected cells linking to pathogenicity and immune evasion[10–14]. Recently, xrRNAs were also found to spread in several members of the plant virus genus *Dianthovirus*, where they are associated with both noncoding and protein-coding regions of the viral genomes[9,15,16]. Despite the divergence in primary sequences, crystal structures of xrRNAs from the flaviviruses and plant-infecting viruses reveal a conserved ring-like 3D fold that creates a protective ring around the 5′-end of the RNA structures[15–19]. The fold of xrRNAs is sufficient for their exoribonuclease resistance ability, and no accessory proteins or chemical modifications of the RNAs are required. Structural and biochemical analysis suggests a "molecular brace" model for xrRNAs' high resistance to directional degradation by the 5′→3′ exoribonucleases[17]. Because of the robust 5′→3′ exoribonuclease resistance, xrRNAs have been applied to visualize the mRNA degradation intermediates and to monitor the spatiotemporal mRNA dynamics in vivo[20,21]. Furthermore, the mechanical anisotropy of ZIKV-xrRNA1, which responds to mechanical stretching in a direction-dependent manner, has recently been confirmed by nanopore sensing technique, suggesting potential biomedical applications of xrRNAs as key elements to building RNA-based biomaterials with controllable mechanical anisotropy[22].

Based on different secondary structure patterns, flaviviral xrRNAs are grouped into two distinct classes[23]. Recently, the class 1 xrRNAs are further divided into two subclasses 1a and 1b, according to the divergence in sets of tertiary interactions that form the conserved ring-like fold[24]. The subclass 1a xrRNAs predominantly exist in MBFVs in tandem (xrRNA1 and xrRNA2)[25], whose ring-like 3D structure is formed through specific structural motifs, including a three-way junction formed by P1, P2, and P3 and an additional P4 helix, two interwoven pseudoknots (PK1, PK2) and other long-range tertiary interactions, such as complex base-stacking arrangements and base triples[17,18,26] (Fig. 1a). Examining a large set of subclass 1a xrRNAs reveals highly conserved sequences, including a conserved cytosine between P2 and P3 (J2/3) and tertiary interaction patterns but also some variability (Supplementary Fig. 1a). Notably, the PK1 length remains almost identical (1–2 bp) but the PK2 length varies from 2 bp to 7 bp across different MBFVs[26] (Fig. 1a, Supplementary Fig. 1a). Previous structural and biochemical studies suggest that PK2 of xrRNAs is conformationally dynamic or transiently formed[17,18,27]. Murray Valley Encephalitis virus (MVE) xrRNA2 and ZIKV-xrRNA1, in which the PK2

lengths are 3 bp and 4 bp, respectively, adopt partially folded (PK2 is not formed) and fully folded (PK2 is formed) conformations in the respective crystal structures[17,18] (Fig. 1b). Chemical probing experiments show higher chemical reactivity for DENV2 xrRNA with shorter PK2, in which PK2 lengths of DENV2-xrRNA1 and xrRNA2 are 2 bp and 4 bp, respectively[27]. In addition, previous in vitro Xrn1 resistance and cellular sfRNA formation assays revealed that PK1 (G3C), J2/3, and PK2 mutations in various xrRNAs severely decrease the Xrn1 resistance of xrRNAs and affect sfRNA formation in vivo[11,17,18,27]. As PK1, PK2, and J2/3 in xrRNAs are critical for maintaining the ring-like 3D architecture and/or mechanical anisotropy[18,22], how PK2 length variation and these tertiary motifs modulate xrRNAs's folding, conformational dynamics and the robustness of Xrn1 resistance is intriguing.

In this study, we first used small-angle X-ray scattering (SAXS) and Xrn1 resistance assay to investigate how PK2 length variation of a set of 11 flaviviral xrRNAs modulates their folding and robustness of Xrn1 resistance. Next, we characterized the folding dynamics and energy landscapes of xrRNAs from DENV2, ZIKV, and WNV using single-molecule Fluorescence Resonance Energy Transfer (smFRET). The PK2 lengths among these xrRNAs are 2 bp, 4 bp, and 6 bp, respectively, and site-specific fluorescent labeling of these RNAs was achieved using the NaM-TPT3 unnatural base pair (UBP) system. Our data demonstrate a strong correlation among xrRNAs' folding, conformational dynamics, and robustness of Xrn1 resistance, which are highly $Mg^{2+}$-dependent, and the $Mg^{2+}$-dependence is modulated by the PK2 length. xrRNAs with longer PK2 (>5 bp) require less $Mg^{2+}$ to stabilize the native fold and to block the degradation by Xrn1, thus exhibiting reduced conformational dynamics even at low $Mg^{2+}$. Moreover, xrRNAs with longer PK2 can tolerate mutations at key tertiary motifs such as PK1 or J2/3 at high $Mg^{2+}$, which causes destructive effects to xrRNAs with shorter PK2, suggesting $Mg^{2+}$-dependent high cooperativity among key tertiary interactions. Collectively, these data underline the importance of long-range pseudoknot interactions in modulating the folding, conformational dynamics, and robustness of Xrn1 resistance of flaviviral xrRNAs.

## Results

**PK2 length variation modulates $Mg^{2+}$-dependence of xrRNAs' folding and robustness of Xrn1 resistance.** $Mg^{2+}$ has been shown to induce a structural transition and to stabilize the compact ring-like 3D architecture of the ZIKV-xrRNA1[22]. To test how PK2 length variation affects the folding of flaviviral xrRNAs, SAXS measurements were performed for a set of 11 flaviviral xrRNAs in the presence of 5 mM $Mg^{2+}$ or 5 mM EDTA, respectively. These xrRNAs are highly conserved in primary sequences, secondary structures, as well as tertiary structures[24], whereas their PK2 lengths vary from 2 bp to 7 bp (Supplementary Fig. 1). The scattering profiles, with scattering intensity $I(q)$ plotted against momentum transfer $q$, the pair distance distribution function (PDDFs), and the dimensionless Kratky plots transformed from the scattering profiles for the respective xrRNAs are shown in Supplementary Fig. 2a–c. All flaviviral xrRNAs in 5 mM $Mg^{2+}$ are highly folded, characterized by the bell-shaped peaks in the dimensionless Krakty plots, by contrast, the folding behaviors in 5 mM EDTA vary along with the PK2 lengths of xrRNAs (Fig. 1c). xrRNAs with short PK2 (<5 bp) exhibit partially folded characteristics in 5 mM EDTA, but xrRNAs with long PK2 (>5 bp) are better folded in 5 mM EDTA (Fig. 1c), though less compact than that in 5 mM $Mg^{2+}$ likely due to transient unfolding and unstable tertiary interactions in the absence of $Mg^{2+}$ (Supplementary Fig. 2c). Plotting the $R_g$ and

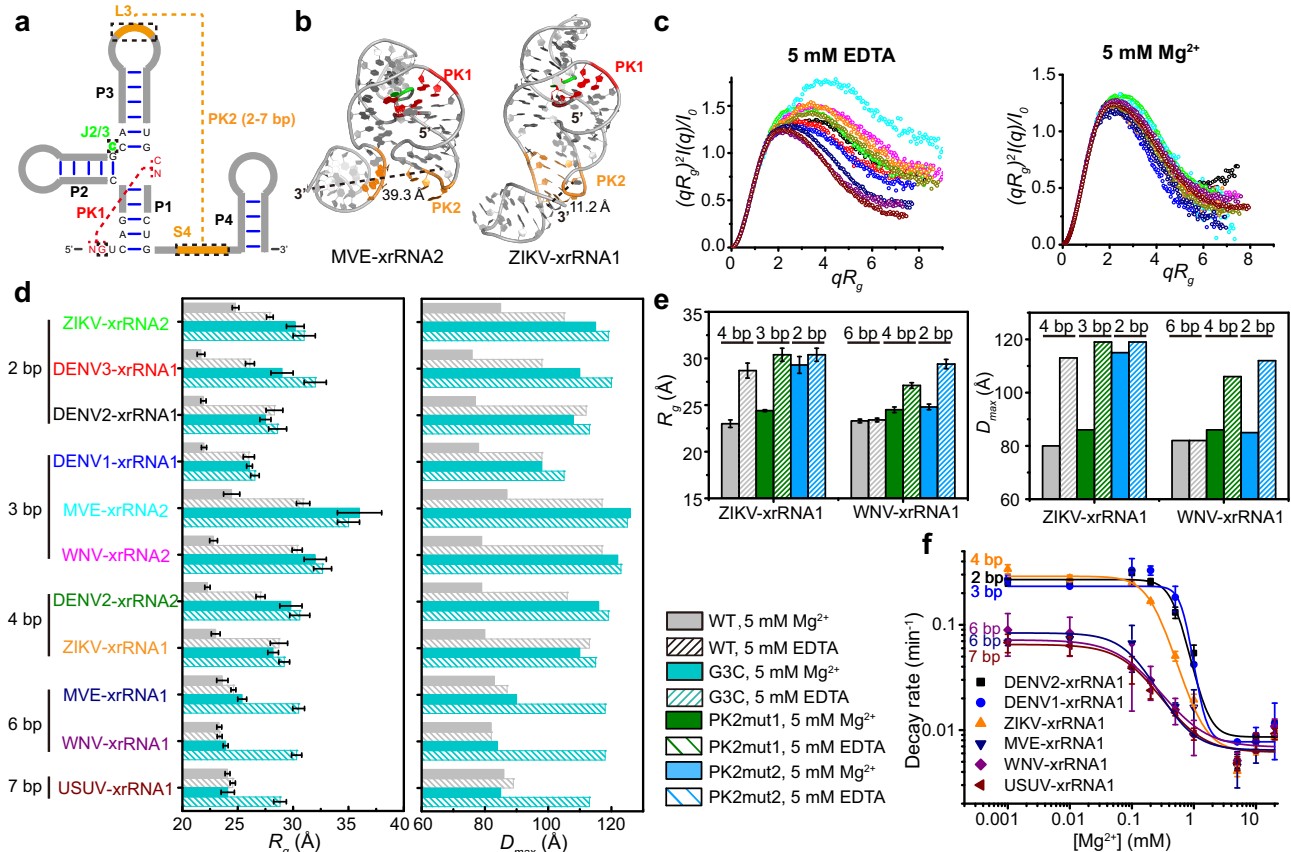

**Fig. 1 Mg$^{2+}$-dependence of flaviviral xrRNAs' folding and robustness of Xrn1 resistance is modulated by PK2 length variations. a** Schematic representation of the secondary structure for flaviviral subclass 1a xrRNAs. The red and orange lines indicate the PK1 and PK2 pseudoknot interactions, respectively. The conserved nucleotide C between P2 and P3 (J2/3) is colored in green. $N$ represents any nucleotides. Sites for mutations to disrupt the key tertiary interaction motifs are indicated with the dashed boxes. **b** Crystal structures of MVE-xrRNA2 (PDB ID: 4PQV) and ZIKV-xrRNA1 (PDB ID: 5TPY). The color codes are the same as that in **a**. The distances between residues corresponding to the labeling sites in L3 and the 3′-end of the dye-labeled xrRNAs are measured. **c** The dimensionless Kratky plots for a set of 11 xrRNAs in 5 mM Mg$^{2+}$ or EDTA. **d–e** $R_g$ and $D_{max}$ for the WT and G3C mutants of a set of 11 xrRNAs (**d**) and PK2 mutants of ZIKV- and WNV- xrRNA1 (**e**). The color codes for the respective xrRNAs in **c–d** are the same. Each column in **d–e** represents an independent experiment ($n = 1$) and the error bars are propagated uncertainties calculated by GNOM. **f** Relative Xrn1 decay rates for xrRNA1s *versus* Mg$^{2+}$ concentrations. The decay rates were calculated by fitting the fluorescence traces (see supplementary Fig. 4e–j) with a single exponential curve. Data are presented as mean values ± SEM from three independent experiments. Source data for **c** and **f** are provided as a Source Data file.

$D_{max}$ parameters derived from PDDFs along with xrRNAs' PK2 length show that, $R_g$ and $D_{max}$ in 5 mM Mg$^{2+}$ are significantly smaller than that in 5 mM EDTA for xrRNAs with short PK2 (<5 bp), but are only slightly smaller for xrRNAs with long PK2 (>5 bp) (Fig. 1d–e, Supplementary Fig. 2b, and Table S3). ab initio shape reconstruction was performed for xrRNAs in the presence of EDTA and Mg$^{2+}$. The resulting shape envelopes for all xrRNAs in Mg$^{2+}$ are compact, whereas the shape envelopes for xrRNAs with short PK2 (<5 bp) in EDTA are relatively elongated, the shape envelopes for xrRNAs with long PK2 (>5 bp) in EDTA are close to that in Mg$^{2+}$ (Supplementary Fig. 3). Clearly, these results suggest coherent structural collapse upon Mg$^{2+}$ binding for all xrRNAs. Furthermore, the Mg$^{2+}$-dependence of xrRNAs' folding is modulated by PK2 length variation, thus the folding of xrRNAs with long PK2 is less dependent on Mg$^{2+}$.

The primary biological function of xrRNAs is to resist degradation by 5′→3′ exoribonuclease such as Xrn1. To understand how PK2 length variation modulates the robustness of xrRNAs' Xrn1 resistance, a fluorescence-based Xrn1 decay assay[27] was performed for flaviviral xrRNAs with different PK2 lengths at various Mg$^{2+}$ concentrations ranging from 0.001 to 20 mM (Supplementary Fig. 4). As Mg$^{2+}$ is essential for the

optimal activity of Xrn1[28,29], to correct the effect of Mg$^{2+}$ on Xrn1 activity, the malachite green (MG) aptamer alone is used as the control to evaluate the Mg$^{2+}$-dependence of exoribonuclease activity (Supplementary Fig. 4b–d). The decay rates of the respective flaviviral xrRNAs were then normalized by the decay rates of the MG aptamer at the corresponding Mg$^{2+}$ concentrations (Fig. 1f). All flaviviral xrRNAs exhibit strong robustness of Xrn1 resistance in high Mg$^{2+}$, but their robustness differs significantly at lower Mg$^{2+}$ concentrations (0.1–1 mM). Although xrRNAs with long PK2 (>5 bp) display strong robustness of Xrn1 resistance even in 0.1 mM Mg$^{2+}$, the robustness of Xrn1 resistance for xrRNAs with short PK2 (<5 bp) decreases by 10–30 times (Fig. 1f). Obviously, there is a strong correlation between the Mg$^{2+}$-dependence of the robustness of Xrn1 resistance and xrRNAs' PK2 length, and thus the robustness of Xrn1 resistance for xrRNAs with short PK2 are more sensitive to Mg$^{2+}$ concentration changes than the ones with long PK2 (Fig. 1f).

To further support the correlation of PK2 length with Mg$^{2+}$-dependence of xrRNAs' folding and robustness of Xrn1 resistance, PK2 mutants of both ZIKV-xrRNA1 (ZIKV-PK2mut1, ZIKV-PK2mut2) and WNV-xrRNA1 (WNV-PK2mut1, WNV-

PK2mut2) were generated, whose PK2 lengths are shortened from 4 bp to 3 bp, 2 bp, and from 6 bp to 4 bp, 2 bp, respectively (Supplementary Fig. 5a). Their folding and Xrn1 resistance activity were examined by SAXS and Xrn1 decay kinetics assay (Fig. 1e, Supplementary Fig. 5b–d). For PK2 mutants of ZIKV-xrRNA1, as PK2 lengths (<5 bp) decrease, the RNA variants become less compact in 5 mM $Mg^{2+}$ inferred from the dimensionless Kratky plots. It should be noted that while ZIKV-PK2mut1 can fold similarly (though less compact) as the wild type in 5 mM $Mg^{2+}$. ZIKV-PK2mut2 variant, whose PK2 length is expected to be 2 bp, is unfolded in 5 mM $Mg^{2+}$. By contrast, some wild-type xrRNAs whose PK2s are 2 bp (Supplementary Fig. 1b) are well folded in 5 mM $Mg^{2+}$ (Fig. 1c). It is likely that decreasing PK2 length from 4 bp to 2 bp in ZIKV-xrRNA1 severely disrupts the cooperative interaction networks involving PK2 that guide RNA folding, resulting in the unfolded ZIKV-PK2mut2 in 5 mM $Mg^{2+}$. Since their PK2 lengths are shorter than 5 bp, similarly, the folding of ZIKV-xrRNA1 and ZIKV-PK2mut1 are strongly dependent on $Mg^{2+}$. For PK2 mutants of WNV-xrRNA1, as PK2 lengths decrease, the RNA variants become less compact, however, both WNV-PK2mut1 and WNV-PK2mut2 fold closely to the wild type in 5 mM $Mg^{2+}$. Furthermore, the folding of WNV-PK2mut2 (PK2: 2 bp) is more sensitive to $Mg^{2+}$ than that of WNV-PK2mut1 (PK2: 4 bp) and WNV-WT (PK2: 6 bp). Such observations were also supported by $R_g$ and $D_{max}$ parameters derived from the respective PDDFs (Fig. 1e, Table S3). The Xrn1 decay assays on PK2 mutants of ZIKV- and WNV- xrRNA1s at various $Mg^{2+}$ concentrations suggest that xrRNAs with shorter PK2 require higher $Mg^{2+}$ to resist the degradation by Xrn1. Consistent with its folding property, ZIKV-PK2mut2 exhibits poor Xrn1 resistance activity in 5 mM $Mg^{2+}$, but the Xrn1 resistance activity significantly enhances as $Mg^{2+}$ concentration increases to 20 mM (Supplementary Fig. 5d).

**Site-specific fluorescent labeling of xrRNAs using UBP system.** The cellular functions of RNA are governed by its specific 3D structure and structural dynamics[5]. To understand how PK2-length variation modulates xrRNAs' conformational dynamics and their Xrn1 resistance, we intend to apply smFRET measurements based on total internal reflection fluorescence (TIRF) microscopy[30] (Fig. 2a) to reveal the folding energy landscapes of xrRNA1 from DENV2, ZIKV, and WNV, whose PK2 lengths are 2 bp, 4 bp, and 6 bp, respectively, thus representing xrRNAs with PK2 length from short to long (Supplementary Fig. 6a–c). FRET pairs (Cy3 and Cy5) were introduced into these xrRNAs at suitable locations to ensure that xrRNAs' folding would result in evident FRET efficiency changes. The 3′ ends of xrRNAs were co-transcribed with extended single-stranded RNA sequences, which were annealed to a complementary DNA oligomer containing 5′-end biotin and a 3′-end Cy3 for surface immobilization and fluorophore labeling (Fig. 2a, Supplementary Fig. 6a–c). Based on their secondary and tertiary structures, residues C27 in DENV2-xrRNA1, U29 in ZIKV-xrRNA1, and A30 in WNV-xrRNA1 locate in Loop 3 adjacent to PK2 pseudoknots were selected to label with Cy5 to generate FRET signals with Cy3-labeled DNA annealed at their 3′ ends[18,31] (Supplementary Fig. 6a–c). Site-specific internal Cy5 labeling of xrRNAs was achieved using the UBP system containing NaM and TPT3 originally developed by Romesberg's group[32,33] (Fig. 2b), which were recently utilized to develop strategies for post-transcriptionally site-specific labeling of large RNAs with spin or gold nanoparticle[34,35]. Using a similar strategy[35], DNA templates containing an upstream T7 promoter and dNaM modification at the template strands were prepared by overlapping PCR reactions, the selected residues in the respective

xrRNAs were then replaced with alkyne-modified $rTPT3^{CO}$ by in vitro transcription (Fig. 2c), allowing azide-modified Cy5 to be coupled with the alkyne group of $rTPT3^{CO}$ in the transcripts through click chemistry reaction (Fig. 2d). The effects of UBP-based fluorescent labeling on the respective xrRNAs were assayed, as shown in Supplementary Fig. 6d–f. Their Xrn1 resistance activities in high $Mg^{2+}$ concentrations are minimally perturbed, thus the UBP method presents an efficient site-specific fluorescence labeling strategy for large RNAs.

**PK2 length variation modulates $Mg^{2+}$-dependent conformational ensembles of xrRNAs.** The time-dependent Cy3 and Cy5 fluorescence signals emitted from the respective dye-labeled xrRNAs were collected over a wide range of $Mg^{2+}$ concentrations (0.001−100 mM) (Fig. 3a–c). The FRET efficiency ($E_{FRET}$) was calculated from the Cy3 and Cy5 intensities of thousands of individual molecules, which reflects the intramolecular distance between Cy3 and Cy5 labeling sites, thus corresponding to the extent of tertiary folding[36] (Fig. 2a). Over the whole range of $Mg^{2+}$ concentrations, the FRET histograms of all three xrRNAs displayed three FRET states (Fig. 3d–f, Supplementary Fig. 7a–c), whose peak centers and populations were extracted via Gaussian fitting. The mean $E_{FRET}$ values of the low-FRET state ($L_{state}$), the intermediate-FRET state ($I_{state}$) and the high-FRET state ($H_{state}$) are ~0.3, ~0.5, and ~0.9, respectively. In the crystal structures of MVE-xrRNA2 and ZIKV-xrRNA1, representing the partially folded and folded conformations of xrRNAs[10,11], the distances between residues corresponding to the labeling sites in L3 and the 3′ end of the dye-labeled xrRNAs are estimated as ~39 Å and ~11 Å, respectively (Fig. 1b), which should generate intermediate and high-FRET efficiencies, respectively. Thus, the $L_{state}$, $I_{state}$, and $H_{state}$ are assigned as the unfolded, partially folded, and folded states, respectively (Fig. 2a), and all xrRNAs display spontaneous transitions among these three conformational states (Fig. 3a–c).

At low $Mg^{2+}$ concentration (0.001 mM), DENV2-xrRNA1 and ZIKV-xrRNA1 are mostly in the $L_{state}$ corresponding to the unfolded conformation and occasionally sample the $I_{state}$ (Fig. 3d, e, Supplementary Fig. 7a, b). By contrast, WNV-xrRNA1 spontaneously transits among the $L_{state}$, $I_{state}$, and $H_{state}$ with similar populations (Fig. 3f, Supplementary Fig. 7c). As $Mg^{2+}$ concentration increases, xrRNAs sample the $H_{state}$ more frequently for longer time periods. At 1 mM $Mg^{2+}$, although DENV2-xrRNA1 still samples mostly the $L_{state}$, WNV-xrRNA1 samples mostly the $H_{state}$. At above 5 mM $Mg^{2+}$, the $H_{state}$ becomes the most dominant population for all xrRNAs, indicating that all xrRNAs greatly favor the folded conformation, consistent with the SAXS experiments (Fig. 1c). Surprisingly, even at 100 mM $Mg^{2+}$, which is many times higher than the physiological concentration, all the xrRNAs continue to sample the $L_{state}$ and $I_{state}$, indicating their inherent structural dynamics. Transition density plot (TDP) analysis clearly reveals that xrRNAs mainly transit directly between $L_{state}$ and $I_{state}$, and between $I_{state}$ and $H_{state}$ (Fig. 3g–i, Supplementary Fig 7d–f). As $Mg^{2+}$ increases, the number of molecules that transit between different FRET states decreases, implying that xrRNAs become more stable.

Although all three xrRNAs display conformational heterogeneity and dynamic transitions among different conformational states, their dependences on $Mg^{2+}$ differ significantly. The fraction of each state in the respective xrRNAs was calculated and plotted against $Mg^{2+}$ concentrations (Fig. 3j–l, Supplementary Table S4). The decrease of total occupancies of $L_{state}$ is accompanied by the increase of $H_{state}$ for all three xrRNAs as $Mg^{2+}$ concentrations increase. By contrast, the total occupancies

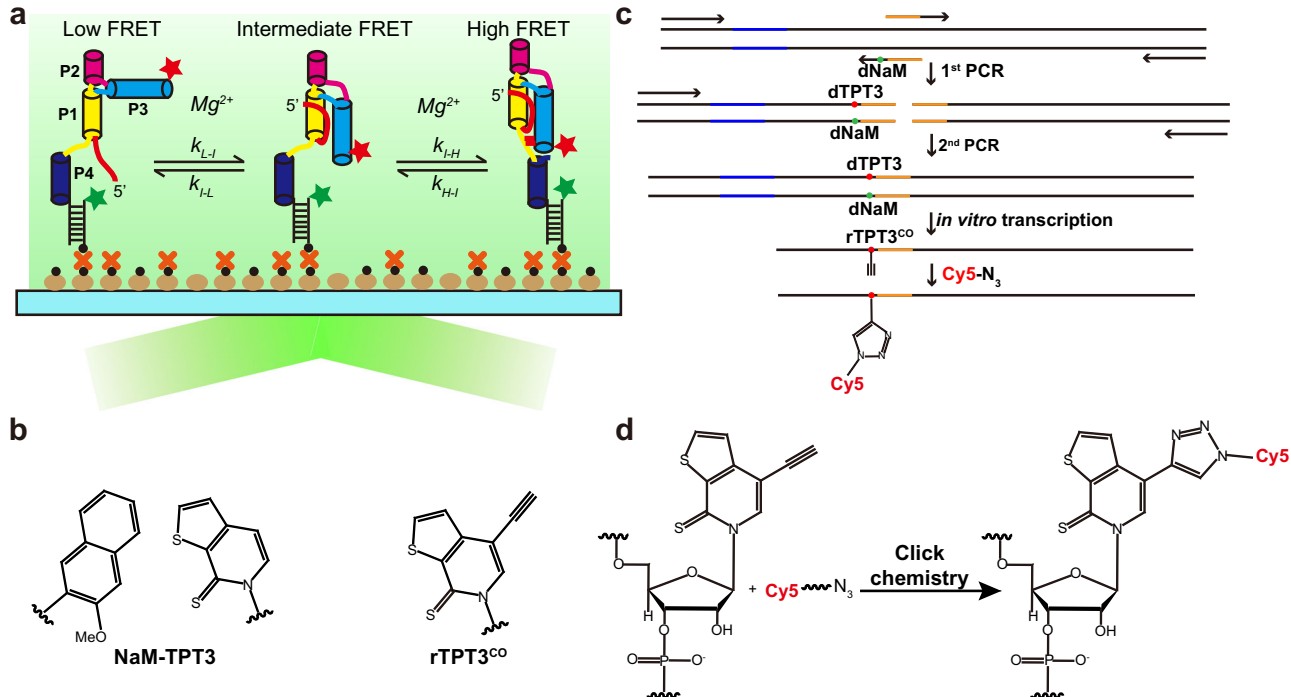

**Fig. 2 UBP-based strategy for RNA fluorescent labeling facilitates TIRF-based smFRET measurements of flaviviral xrRNAs. a** Schematic representation of the TIRF-based smFRET analysis of the folding of flaviviral xrRNAs. The low-FRET ($L_{state}$), intermediate-FRET ($I_{state}$), and high-FRET ($H_{state}$) states correspond to the unfolded, partially folded, and folded states of xrRNA1, respectively. **b** Chemical structures of the parental NaM-TPT3 unnatural base pair and alkyne-modified TPT3 (rTPT3$^{CO}$). **c** DNA templates of xrRNAs containing T7 promoter (shown in blue) and dNaM modification at the template strands were prepared by two-steps overlapping PCR, which was followed by in vitro transcription using rNTP mix supplemented with rTPT3$^{CO}$TP, allowing site-specific labeling of purified transcripts with azide-modified Cy5 dye. **d** Conjugation of azide-modified Cy5 with TPT3$^{CO}$-modified RNA via Click chemistry.

of $I_{state}$ remain almost constant over the full range of $Mg^{2+}$ concentrations for DENV2-xrRNA1 and ZIKV-xrRNA1, but the total occupancy of $I_{state}$ decreases as $Mg^{2+}$ increases for WNV-xrRNA1 (Fig. 3j–l). The critical $Mg^{2+}$ concentrations to stabilize xrRNA structure, defined as the concentrations above which the fraction of $H_{state}$ is higher than that of $L_{state}$, are 7 mM, 1.5 mM, and 0.2 mM for DENV2-xrRNA1, ZIKV-xrRNA1, and WNV-xrRNA1, respectively. Clearly, xrRNAs with longer PK2 require less $Mg^{2+}$ to stabilize the folded conformations.

To analyze the correlation between xrRNAs' conformational dynamics and Xrn1 resistance activity across a broad range of $Mg^{2+}$ concentrations, the Xrn1 decay rates are plotted against the populations of $H_{state}$ at the same $Mg^{2+}$ concentration (Fig. 3m). Clearly, for all three xrRNAs, high $Mg^{2+}$ leads to a slow Xrn1 decay rate and large $H_{state}$ population, which is supported by the strong negative correlation between them through a global analysis across the three xrRNAs. Thus, the $H_{state}$, corresponding to the native folded xrRNA conformation, confers Xrn1 resistance.

**PK2 length variation modulates $Mg^{2+}$-dependent conformational dynamics of xrRNAs.** To better understand how PK2 length variations modulate conformational dynamics of xrRNAs, the dwell time distributions of all three states and transition rates among them at each $Mg^{2+}$ concentration were extracted from individual single-molecule trajectories according to previously reported procedures (Fig. 4a–c, Supplementary Table S4)[37,38]. In general, the transition rates of $H_{state} \rightarrow I_{state}$ and $I_{state} \rightarrow L_{state}$ decrease significantly when $Mg^{2+}$ concentration increases; whereas the rates of their reverse reactions $I_{state} \rightarrow H_{state}$ and $L_{state} \rightarrow I_{state}$ are less sensitive to changes of $Mg^{2+}$ concentrations.

Although $H_{state}$ can directly transit to $L_{state}$ under almost all conditions, its reverse reaction, direct transition from $L_{state}$ to $H_{state}$, is relatively slow and rarely seen, even at high $Mg^{2+}$ when $H_{state}$ is the most stable state. Furthermore, the transition density plots for the three xrRNAs in different $Mg^{2+}$ concentrations also demonstrate that xrRNAs can rarely transit between $L_{state}$ and $H_{state}$ (Fig. 3g–i, Supplementary Fig. 7d–f). These phenomena strongly support that the folding of xrRNAs follows a definite sequential pathway corresponding to transitions from $L_{state}$ to $I_{state}$, then to $H_{state}$, consistent with the previously proposed folding model of xrRNA that the formation of 5'-end PK1 causes P1 and P3 to swing into proper position, resulting in the formation of PK2 by L3 and S4 (Fig. 2a)[26,39]. At last, the transition rates among three FRET states in WNV-xrRNA1 are smaller than that in DENV2-xrRNA1 and ZIKV-xrRNA1, thus WNV-xrRNA1 containing long PK2 exhibits reduced conformational dynamics.

To better illustrate how $Mg^{2+}$ and PK2-length variation modulates xrRNA's folding and conformational dynamics, the folding free-energy landscapes of each xrRNA were estimated from transition rates among unfolded, partially folded, and folded states (Fig. 4d–f). The $L_{state}$ was set as the ground state ($\Delta G = 0$), with which free energies ($\Delta G$) of $I_{state}$ and $H_{state}$ were calculated (see Methods). Because direct transition events from $L_{state}$ to $H_{state}$ rarely occur and the rate of $L_{state} \rightarrow H_{state}$ was less well determined, we did not utilize this rate to estimate the energy barrier between $L_{state}$ and $H_{state}$. Clearly, $Mg^{2+}$ significantly stabilizes both $I_{state}$ and $H_{state}$ for all xrRNAs. For example, when $Mg^{2+}$ increases from 0.001 mM to 1 mM, the $I_{state}$ and $H_{state}$ of DENV2-xrRNA1 is stabilized by ~0.24 kcal/mol and ~1.3 kcal/mol, respectively, accompanied by the decrease of transition energy barrier from $I_{state}$ to $H_{state}$ by ~0.53 kcal/mol and increase

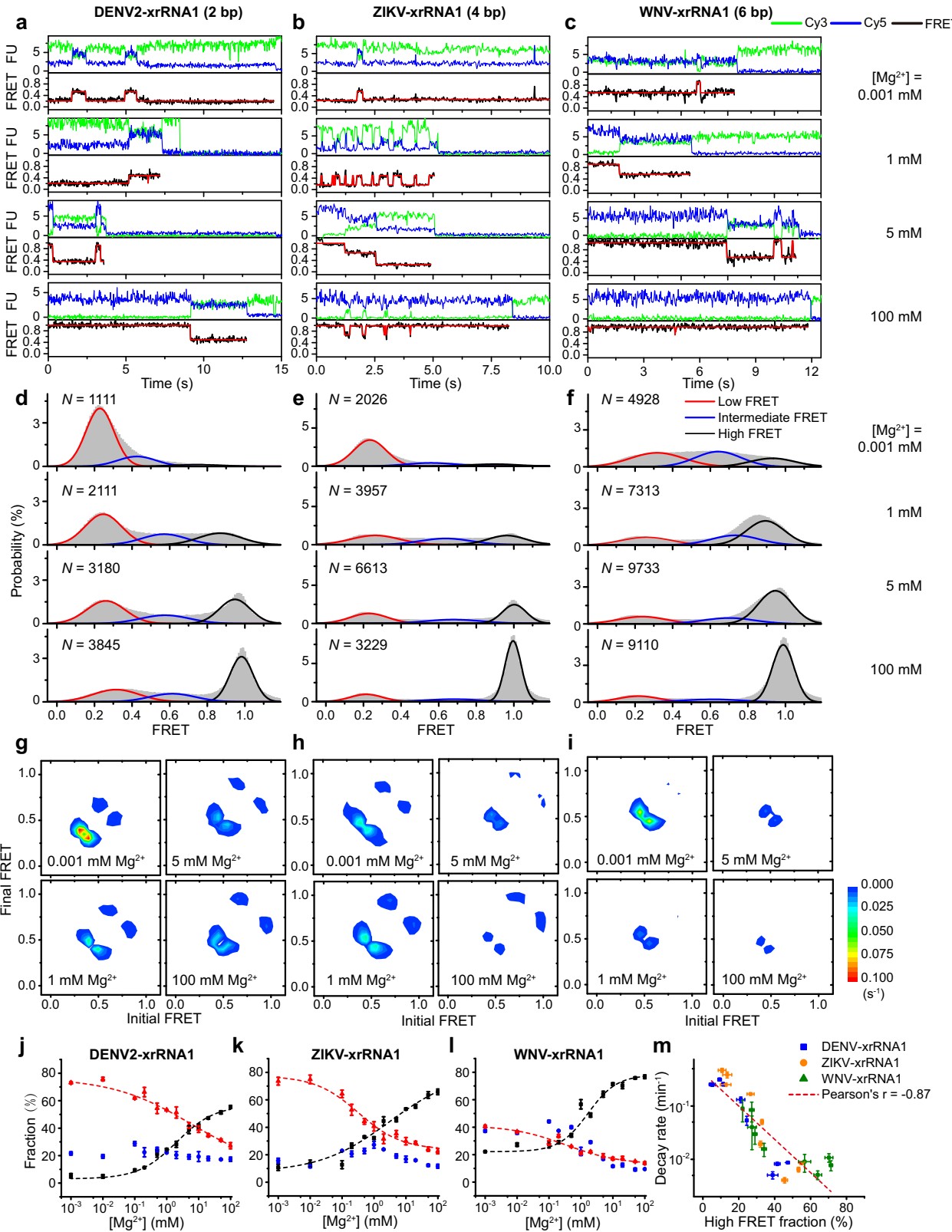

of transition energy barrier from $H_{state}$ to $I_{state}$ by ~0.77 kcal/mol (Fig. 4d). Similar stabilizing effects of $Mg^{2+}$ were also observed for ZIKV-xrRNA1 and WNV-xrRNA1, though the effects of $Mg^{2+}$ were less evident to WNV-xrRNA1 (Fig. 4e, f). These energy landscapes further emphasize that the major contributing factor to stabilize folded state at high $Mg^{2+}$ is to increase energy

barrier of $H_{state} \rightarrow I_{state}$ to decrease transition rate of $H_{state} \rightarrow I_{state}$, whereas the moderately decreased energy barrier of $I_{state} \rightarrow H_{state}$ causing increased transition rate of $I_{state} \rightarrow H_{state}$, can only be accounted as the minor stabilizing factor of high $Mg^{2+}$. The PK2 length strongly affects the stability of $H_{state}$ across the xrRNAs, especially at the low $Mg^{2+}$. Besides, the transition

**Fig. 3 $Mg^{2+}$-dependent folding dynamics of flaviviral xrRNAs by smFRET. a–c** Representative smFRET traces for DENV2-xrRNA1 (**a**), ZIKV-xrRNA1 (**b**), and WNV-xrRNA1 (**c**) in different $Mg^{2+}$ concentrations ranging from 0.001 to 100 mM. The idealized FRET traces generated by the hidden Markov model (red) were overlaid on the experimental FRET traces (black). **d–f** FRET histograms of DENV2-xrRNA1 (**d**), ZIKV-xrRNA1 (**e**), and WNV-xrRNA1 (**f**) at various $Mg^{2+}$ concentrations. $N$ denotes the total number of traces used to generate each histogram from three independent experiments. **g–i** Transition density plots (TDPs) for DENV2-xrRNA1 (**g**), ZIKV-xrRNA1 (**h**), and WNV-xrRNA1 (**i**) in different $Mg^{2+}$ concentrations. TDPs were generated from all smFRET traces from three independent experiments. **j–l** Fractional population of FRET states from three-state fitting to FRET histograms in **d–f** for DENV2-xrRNA1 (**j**), ZIKV-xrRNA1 (**k**), and WNV-xrRNA1 (**l**). $Mg^{2+}$-dependence of fractional populations of high-FRET and low-FRET states were fitted using Hill equation, the fractional population of intermediate-FRET states was not fitted. The individual state is colored like that in **d–f**. **m** A global analysis of the correlation between the Xrn1 decay rates and the fractions of high-FRET at the same $Mg^{2+}$ concentration ranging from 3% to 72% for DENV2-xrRNA1, ZIKV-xrRNA1, and WNV-xrRNA1. Data are presented as mean ± SEM. Error bars are SEM of three independent experiments ($n = 3$) in **m**. Source data are provided as a Source Data file.

energy barrier between $I_{state}$ and $H_{state}$ in WNV-xrRNA1 is the highest, thus resulting in the slowest transition rates, consistent with the reduced conformational dynamics of WNV-xrRNA1.

To understand the correlation between xrRNA conformational dynamics and its function, the transition rates among different FRET states of the three xrRNAs were plotted against the Xrn1 decay rates at different $Mg^{2+}$ concentrations (Fig. 4g–j). While the Xrn1 decay rates are in general positively correlated with the transition rates of $k_{H-I}$ and $k_{I-L}$, indicating that the faster xrRNAs escape from the $H_{state}$ to $I_{state}$, or from $I_{state}$ to $L_{state}$, the easier it for them to be degraded by Xrn1, by contrast, there was no significant correlation between Xrn1 decay rates and the transition rates of $k_{I-H}$ and $k_{L-I}$. In addition, the free energy of $H_{state}$ of all xrRNAs exhibits a significant positive correlation with Xrn1 decay rates (Fig. 4k). The correlation among transition rates, free energies, and decay rates was not perfect, which might be affected by different experimental temperatures (25 °C vs 37 °C) and fluorophore labeling used in smFRET experiments. Nevertheless, these analyses reveal that the degradation of xrRNA by Xrn1 is kinetically controlled by the transition of xrRNA from $H_{state}$ to $I_{state}$, thus by the propensity to unfold its structure. In other words, the Xrn1 resistance activity of xrRNAs highly depends on the conformational dynamics of its ring-like structure, which is regulated by $Mg^{2+}$ and PK2-length variations.

**Long PK2 attenuates the mutational effects to xrRNAs' folding and Xrn1 resistance.** Previous results showed that disruptions of PK1 (e.g., G3C mutant in MVEV-xrRNA2)[17], or J2/3 (C22G mutants in ZIKV-xrRNA1)[18] in xrRNAs abolished their ability to resist Xrn1 degradation, suggesting the importance of such tertiary interactions in stabilizing the active conformations of xrRNAs to dictate Xrn1 resistance. In addition to their individual contributions to stability, tertiary interactions in different regions of the RNA may cooperatively stabilize the native structure when formed simultaneously[40].

To understand how PK2-length variation modulates the cooperativity network, PK1 mutants (G3C) of a set of 11 xrRNAs were constructed (Fig. 1a, Supplementary Fig. 1b) and their effects on xrRNAs' global structures were characterized by SAXS. The scattering profiles, PDDFs, and the dimensionless Kratky plots were compared with the respective xrRNA wild types and shown in Supplementary Fig. 2a–c. As featured in the dimensionless Kratky plots, in 5 mM EDTA, G3C mutants of xrRNAs were fully or partially unfolded, (Fig. 5a, Supplementary Fig. 2c). In 5 mM $Mg^{2+}$, whereas xrRNAs with short PK2 are folded, their G3C mutants were fully or partially unfolded, consistent with a previous study in MVE-xrRNA2 that the PK1 mutant causes a destructive effect[18,22,27]. By contrast, PK1 mutants of xrRNAs with long PK2 such as WNV-xrRNA1 almost exhibit the same folding as the wild types (Fig. 5a, Supplementary Fig. 2c). Thus, the effect of PK1 mutation (G3C) on xrRNAs' folding is modulated by PK2 length, PK1 and PK2

are cooperatively linked to stabilize the folded conformations of xrRNAs, especially in xrRNAs with long PK2. The dependence of PK1 mutational effect to xrRNA structure on PK2 length was also supported by the $R_g$ and $D_{max}$ data derived from the respective PDDFs, the differences in $R_g$ and $D_{max}$ between PK1 mutant and wild type of xrRNAs in $Mg^{2+}$ become less apparent as PK2 length increases (Fig. 1d–e). These results suggest that long PK2 can attenuate the effects of PK1 mutation to xrRNAs' folding in high $Mg^{2+}$, which is however destructive to the folding of xrRNAs with short PK2.

The effect of J2/3 mutation on xrRNAs' structure was also studied by SAXS (Fig. 5b–d, Supplementary Fig. 2d–f), using the J2/3 mutants of DENV2-xrRNA1, ZIKV-xrRNA1, and WNV-xrRNA1 as representatives (Fig. 1a, Supplementary Fig. 8a–c). In 5 mM EDTA, J2/3 mutants of all three xrRNA1s are unfolded. In 5 mM $Mg^{2+}$, although J2/3 mutants of DENV2-xrRNA1 and ZIKV-xrRNA1 become partially folded, J2/3 mutant of WNV-xrRNA1 almost restores the same folded structure as its wild type (Fig. 5b, Supplementary Fig. 2f). Such observations were also supported by the $R_g$ and $D_{max}$ data (Fig. 5c, d). Taken together, long PK2 can also compensate for the destabilizing effects of J2/3 mutation in high $Mg^{2+}$, which is destructive for xrRNAs with short PK2.

To understand how PK2 length variation modulates the mutational responses of xrRNAs to Xrn1 resistance, Xrn1 decay assays were performed for the PK1 and J2/3 mutants of DENV2-xrRNA1, ZIKV-xrRNA1, and WNV-xrRNA1 in various $Mg^{2+}$ concentrations using the respective wild type as control (Fig. 5e±g, Supplementary Fig. 8d–i). While PK1 and J2/3 mutants of DENV2-xrRNA1 and ZIKV-xrRNA1 cannot resist the degradation by Xrn1 at any $Mg^{2+}$ concentrations (Fig. 5e–f, Supplementary Fig. 8d, e, g, h), the decay rates for PK1 and J2/3 mutants of WNV-xrRNA1 decreased significantly with increasing $Mg^{2+}$ concentrations and became close to the decay rates of its WT in 5 mM $Mg^{2+}$ (Fig. 5g, Supplementary Fig. 8f, i), implying that PK1 and J2/3 mutants of WNV-xrRNA1 in 5 mM $Mg^{2+}$ restored their functional native conformations to dictate Xrn1 resistance. As a control, PK2-L3 mutants of DENV2-, ZIKV-, and WNV-xrRNA1s were constructed, in which the nucleotides in L3 involved in PK2 formation are mutated to its complementary nucleotides (Supplementary Fig. 8a–c), thus, the formation of PK2 is completely disrupted in principle. The Xrn1 resistance activities of all PK2-L3 mutants at different $Mg^{2+}$ concentrations were analyzed with the fluorescence-based Xrn1 decay kinetics assay (Fig. 5e–g, Supplementary Fig. 8j–l). Unlike the above PK1 and J2/3 mutants, PK2-L3 mutants of all three xrRNA1s totally abolished the Xrn1 resistance activity at all conditions, even in high $Mg^{2+}$ (5 mM).

**PK1 and J2/3 mutants of xrRNAs with long PK2 populate native conformations in structural ensembles.** To further understand the attenuation effects of long PK2 to mutations of

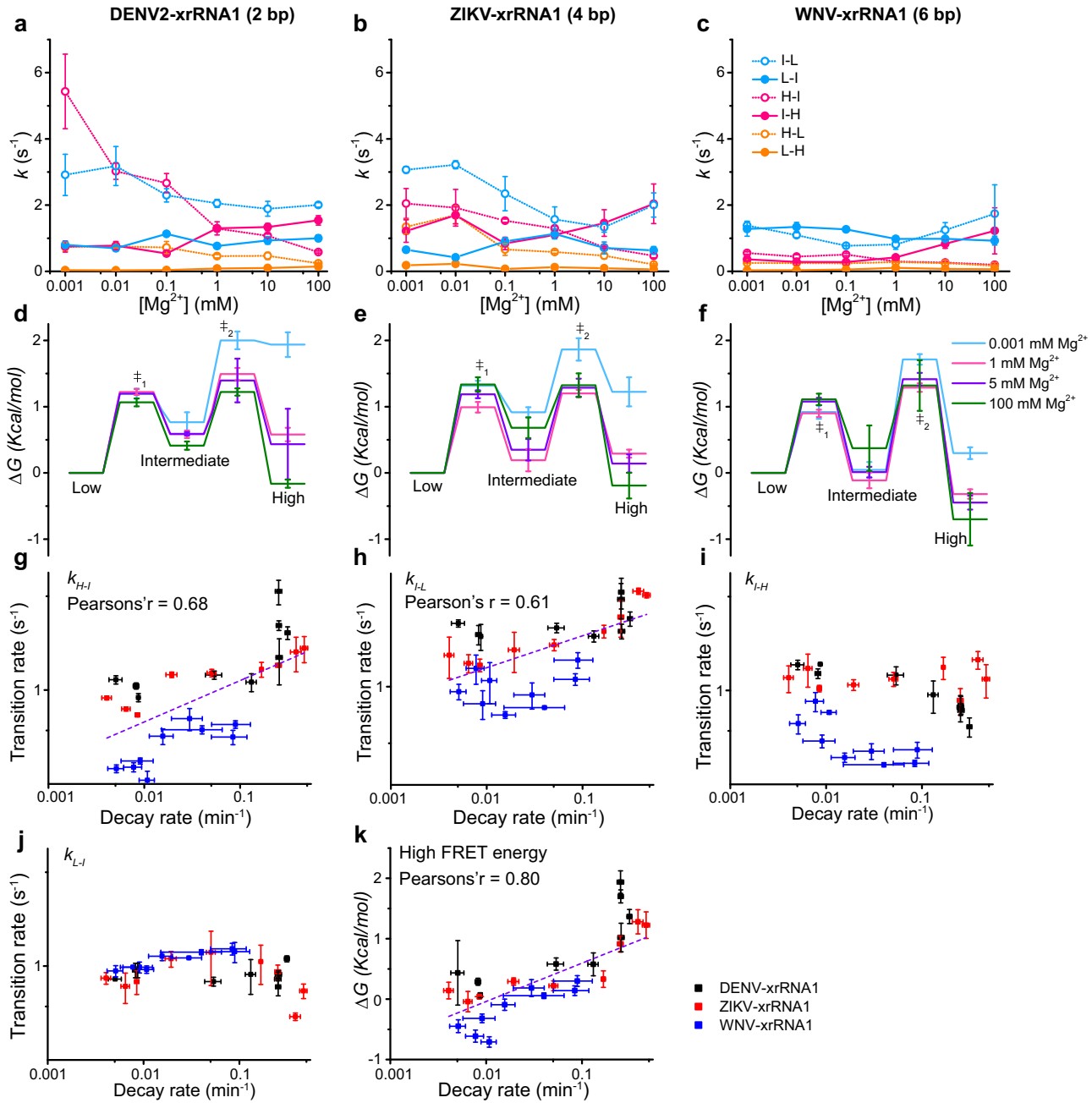

**Fig. 4 Mg²⁺-dependent folding kinetics of flaviviral xrRNAs. a–c** Transition rate constants among the $L_{state}$, $I_{state}$, and $H_{state}$ states for DENV2-xrRNA1 (**a**), ZIKV-xrRNA1 (**b**), and WNV-xrRNA1 (**c**) were plotted against Mg²⁺ concentrations. **d–f** Free-energy diagrams for the Mg²⁺-dependent folding of DENV2-xrRNA1 (**d**), ZIKV-xrRNA1 (**e**) and WNV-xrRNA1 (**f**) at various Mg²⁺ concentrations. **g–j** Correlation analysis of Xrn1 decay rates with the transition rates of $k_{H-I}$, $k_{I-L}$, $k_{I-H}$, and $k_{L-I}$ of xrRNAs at various Mg²⁺ concentrations. **k** Correlation analysis of the free energy of $H_{state}$ with Xrn1 decay rates of xrRNAs at various Mg²⁺ concentrations. Data are presented as mean ± SEM. Error bars are SEM of three independent experiments ($n = 3$) in **a–k**. Source data are provided as a Source Data file.

key tertiary interactions on xrRNAs' folding and Xrn1 resistance in high Mg²⁺, smFRET experiments for the PK1 and J2/3 mutants of DENV2-xrRNA1, ZIKV-xrRNA1, and WNV-xrRNA1 were performed under high Mg²⁺ (10 mM, 5 mM, and 5 mM, respectively). Under such conditions, the respective wild-type xrRNAs were well folded. The labeling schemes for the FRET pairs in the respective mutants were the same as the wild-type xrRNAs (Supplementary Fig. 6a-c).

Similar to the wild-type xrRNAs, the respective PK1 and J2/3 mutants sample unfolded, partially folded, and folded states, suggesting dynamic and heterogeneous structural ensembles

(Fig. 6a–c). For DENV2-xrRNA1 variants in 10 mM Mg²⁺, whereas the wild type samples the $H_{state}$ mostly, the PK1 and J2/3 mutants mostly stay in the $L_{state}$ and transiently sample the $I_{state}$ with no detectable transition to the $H_{state}$ (Fig. 6a, d). For ZIKV-xrRNA1 variants in 5 mM Mg²⁺, besides the $L_{state}$, both PK1 and J2/3 mutants sample the $I_{state}$ and $H_{state}$, but the $H_{state}$ was only rarely populated (Fig. 6b, e). These results were consistent with the complete loss of Xrn1 resistance by the PK1 and J2/3 mutants at 5 mM Mg²⁺ (Fig. 5e, f). By contrast, though the PK1 and J2/3 mutants of WNV-xrRNA1 mainly populate with the $L_{state}$ at 5 mM Mg²⁺, they could frequently transit to $I_{state}$ and $H_{state}$,

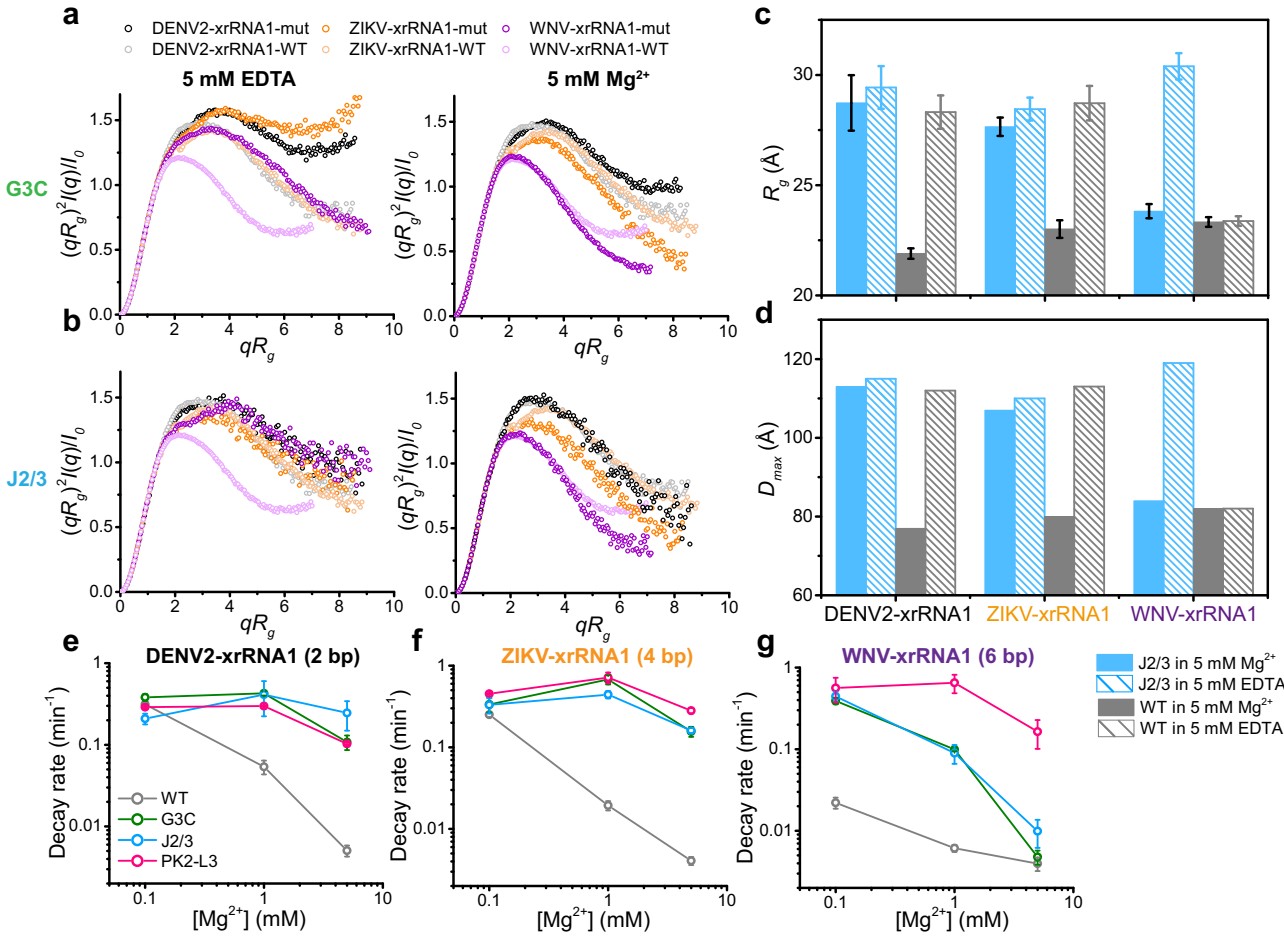

**Fig. 5 Mutations at key tertiary motifs affect the structure and Xrn1 resistance of flaviviral xrRNAs. a–b** The dimensionless kratky plots for G3C (**a**) and J2/3 (**b**) mutants of xrRNA1s from DENV2, ZIKV, and WNV in the presence of 5 mM EDTA (left) or 5 mM $Mg^{2+}$ (right). **c–d** $R_g$ (**c**) and $D_{max}$ (**d**) for the wild type and J2/3 mutants of xrRNA1s from DENV2, ZIKV, and WNV in 5 mM $Mg^{2+}$ or 5 mM EDTA. Each column in **c** represents an independent experiment with $n = 1$ and the error bars are propagated uncertainties calculated by GNOM. **e–g** The relative Xrn1 decay rates for the wild type, PK1, J2/3, and PK2-L3 mutants of DENV2-xrRNA1 (**e**), ZIKV-xrRNA1 (**f**), and WNV-xrRNA1 (**g**) at various $Mg^{2+}$ concentrations. The decay rates were calculated by fitting the fluorescence traces (see supplementary Fig. 8d–l) with a single exponential curve. Data are presented as mean ± SEM. $n = 3$ biologically independent samples examined over three independent experiments in **e–g**. Source data for **a**, **e–g** are provided as a Source Data file.

whose $H_{state}$ fractions could reach 20% and 30% (Fig. 6c, f), respectively. Thus, long PK2 enhances the sampling of native conformations in PK1 and J2/3 mutants of WNV-xrRNA1 in high $Mg^{2+}$ to maintain their Xrn1 resistance (Fig. 5g, Supplementary Fig. 8f, 8i).

smFRET experiments were also performed for the PK2-L3 mutants of DENV2-xrRNA1, ZIKV-xrRNA1, and WNV-xrRNA1, in which the labeling schemes for the FRET pairs are the same as the respective wild type xrRNAs. Though each PK2-L3 mutant of all three xrRNAs samples three conformational states in solution, it mainly populates the intermediate-FRET state and can frequently sample a high-FRET state. However, the FRET efficiencies defined by $E_{FRET}$ values of the intermediate-FRET and high-FRET states are significantly higher and lower than that of the respective WT (Fig. 6d–f, Supplementary Table S5). It is likely that the abolish of PK2 interaction (PK2-L3 mutant) may disrupt the coupling among different tertiary interactions, hence resulting in the nonnative intermediate and misfolded conformation which cannot resist the degradation by Xrn1 even in high $Mg^{2+}$ (Fig. 5e–g).

The sampling of native conformations by mutants of xrRNAs with long PK2 (e.g., WNV-xrRNA1) at high $Mg^{2+}$ was further supported by TDP analysis. As shown in Fig. 6g–i, two contours

corresponding to the interconversions between $I_{state}$ and $H_{state}$ can be observed for the PK1 and J2/3 mutants of WNV-xrRNA1, but not for that of DENV2- and ZIKV- xrRNA1s.

## Discussion

In this study, we combine SAXS, smFRET, and in vitro enzymatic assay to investigate the folding, conformational dynamics, and robustness of Xrn1 resistance upon $Mg^{2+}$ concentration changes and mutations at key tertiary motifs of a set of flaviviral subclass 1a xrRNAs, within which the length of a long-range pseudoknot PK2 varies from 2 bp to 7 bp. We find that xrRNAs' folding, conformational dynamics, and robustness of Xrn1 resistance are highly correlated and affected by $Mg^{2+}$ concentrations, and the $Mg^{2+}$ dependence is modulated by the PK2 length. All xrRNAs sample multiple conformational states in the dynamic ensembles. xrRNAs with long PK2 require less $Mg^{2+}$ to stabilize their native fold and exhibit reduced conformational dynamics and strong robustness of Xrn1 resistance even at a low concentration of $Mg^{2+}$. Furthermore, long PK2 attenuates the mutational effects at key tertiary motifs such as PK1 and J2/3 of xrRNAs through the enhanced sampling of the folded conformation in the respective structural ensembles in high $Mg^{2+}$, which are generally

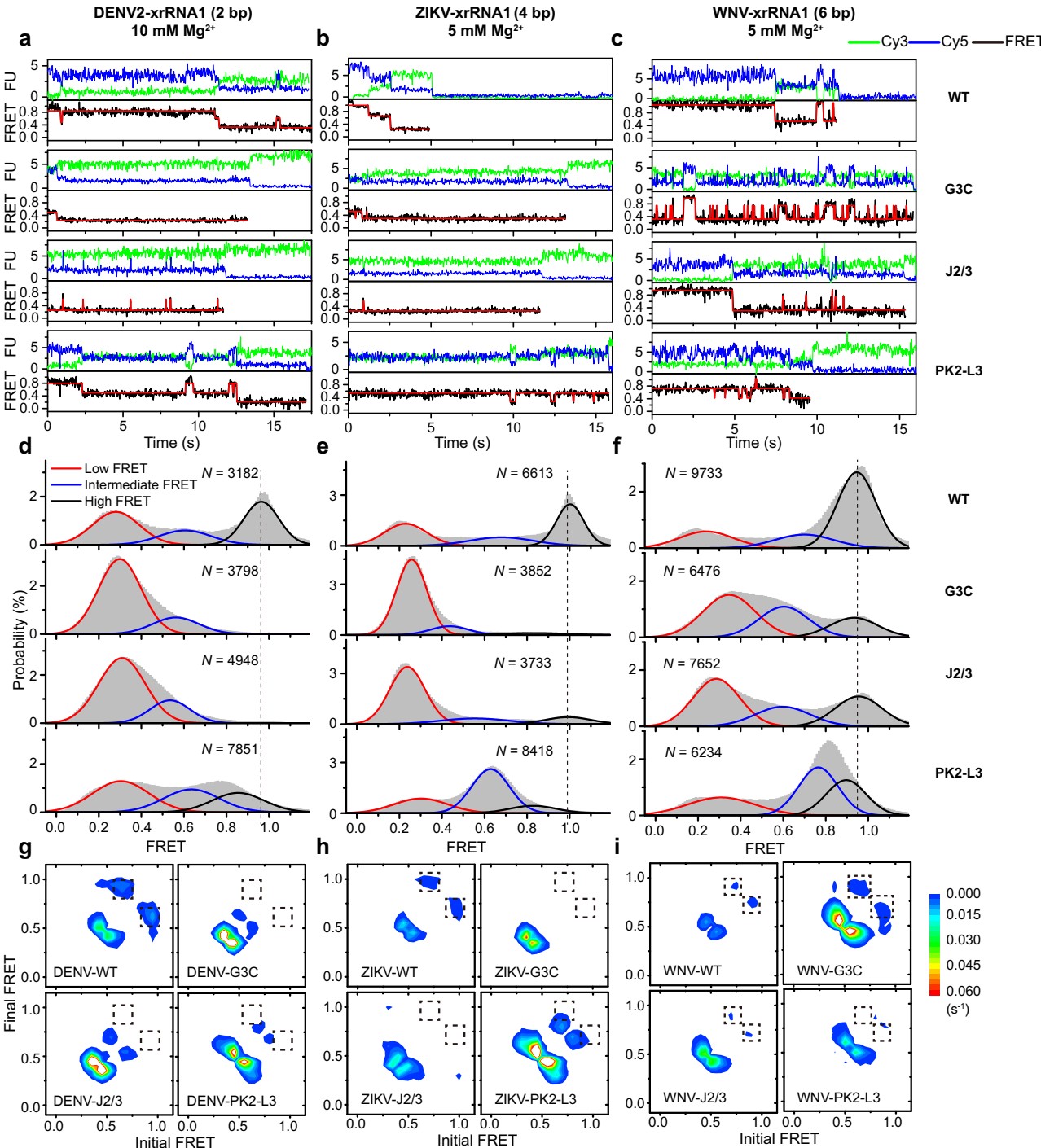

**Fig. 6 Mutations at key tertiary motifs affect the conformational ensembles of flaviviral xrRNAs. a–c** Representative smFRET traces for the wild type, PK1, J2/3 and PK2-L3 mutants of DENV2-xrRNA1 (**a**), ZIKV-xrRNA1 (**b**), and WNV-xrRNA1 (**c**) at 10 mM Mg$^{2+}$, 5 mM Mg$^{2+}$, and 5 mM Mg$^{2+}$, respectively. The idealized FRET traces generated by the hidden Markov model (red) were overlaid on the experimental FRET trace (black). **d–f** FRET histograms for the wild type, PK1, J2/3, and PK2-L3 mutants of DENV2-xrRNA1 (**d**), ZIKV-xrRNA1 (**e**), and WNV-xrRNA1 (**f**) at the corresponding Mg$^{2+}$. $N$ denotes the total number of traces used to generate each histogram from three independent experiments. **g–i** Transition density plots (TDPs) for DENV2-xrRNA1 (**g**), ZIKV-xrRNA1 (**h**), and WNV-xrRNA1 (**i**) mutants. TDPs were generated from all smFRET traces of three independent experiments. The dashed boxes highlight the absence or presence of transitions between $I_{state}$ and $H_{state}$ for xrRNAs mutants. Source data are provided as a Source Data file.

destructive to folding and Xrn1 resistance of xrRNAs with short PK2.

We employed a posttranscriptional labeling strategy based on a UBP system containing NaM-TPT3 to achieve site-specific internal fluorescent labeling of several xrRNAs of 100 nucleotides (nts), which is a prerequisite for smFRET measurements.

Site-specific incorporation of extrinsic fluorophores into large RNAs remains challenging as internal labeling by chemical synthesis is generally limited to RNAs smaller than 80 nts and splint-directed enzymatic ligation is often of low efficiency[41–43]. The NaM-TPT3 UBP-based strategy, which can overcome the size constraints in conventional RNA labeling methods, has

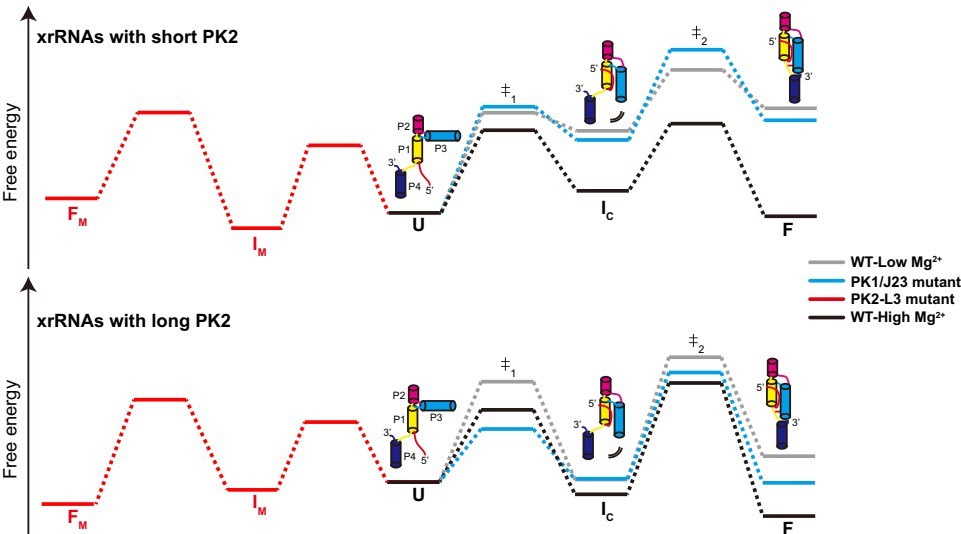

**Fig. 7 The proposed folding free-energy landscape for flaviviral xrRNAs.** The folding is cooperatively sculpted by $Mg^{2+}$ and PK2 length variations. In the absence of $Mg^{2+}$, unfolded state (U) is the most stable conformation. The addition of $Mg^{2+}$ stabilizes the intermediate ($I_C$) and folded state (F). Long PK2 further stabilizes F state and enhances the native state formation. Disruption of tertiary interactions including PK1 or J2/3 destabilizes the $I_C$ and F state significantly, while long PK2 at high $Mg^{2+}$ can compensate for the destabilization effect. However, the complete disruption of PK2 (PK2-L3) results in the formation of nonnative intermediates ($I_M$) and misfolded conformation ($F_M$).

recently been utilized for a site-specific spin and gold nanoparticle labeling to empower the applications of molecular ruler techniques including pulsed electron–electron double resonance spectroscopy (PLEDOR) and X-ray scattering interferometry in large RNAs[34,35]. We expect that the UBP-based site-specific labeling strategy will have broader application in conformational dynamics studies of large RNAs such as Group II introns by smFRET[44,45].

RNA folds through a hierarchical energy landscape, wherein the secondary structure forms first, then tertiary folding occurs in the context of the secondary structure. Recent biophysical studies indicate that ZIKV-xrRNA1 forms alternative secondary and tertiary structures distinct from the ring-like architecture in the absence of $Mg^{2+}$[39], but the presence of $Mg^{2+}$ induces a structural transition to stabilize its compact 3D ring-like structure, thus exhibiting extreme mechanical anisotropy to confer Xrn1 resistance[22]. Our smFRET analysis on three xrRNAs with different PK2 lengths has shown how the energy landscapes of xrRNAs are defined by $Mg^{2+}$ and the PK2-length variations (Fig. 7). All xrRNAs sample three discrete conformational states ($L_{state}$, $I_{state}$, and $H_{state}$), which can be roughly assigned as the unfolded, partially folded, and folded conformations in the dynamic structural ensemble, and PK2 pseudoknot only forms in the folded conformation. $Mg^{2+}$ stabilizes both the $H_{state}$ and $I_{state}$ in all xrRNAs. Furthermore, the $I_{state}$ and $H_{state}$ are more stable for xrRNAs with long PK2 than the ones with short PK2 under the same $Mg^{2+}$ concentrations (such as 5 mM). Thus, high $Mg^{2+}$ and strong PK2 interactions favor the folding of xrRNAs to their native folded conformations, even upon mutations of PK1 or J2/3. Although $H_{state}$ can directly and fully unfold to $L_{state}$, a direct transition from $L_{state}$ to $H_{state}$ without staying in the $I_{state}$ is rarely captured, suggesting cooperative tertiary interactions have to be formed step-by-step in a well-defined order and cannot be formed simultaneously. Taken together, our data support the sequential folding pathway model for flaviviral xrRNAs[26] whose energy landscapes can be cooperatively sculpted by $Mg^{2+}$ and PK2-length variations.

One notable finding in this study is the regulatory role of PK2-length variations in flaviviral xrRNAs. Many RNA viruses utilize

pseudoknots in the control of viral RNA translation and replication[46], pseudoknots have also been found in riboswitches and ribozymes to regulate gene expression[47]. PK1, PK2, and J2/3 are highly conserved tertiary motifs among flaviviral class 1a xrRNAs. Previous results have suggested the significance of PK1, J2/3 tertiary motifs in xrRNAs' structure, in vitro Xrn1 resistance, and cellular sfRNA formation. When formed together, they may cooperatively stabilize the native architecture of xrRNA to dictate its function[40]. Cooperativity is a central feature of RNA folding[48–50]. A cooperative tertiary interaction network has been found to suppress nonnative folding intermediates and guide specific RNA folding[40]. Our study suggests strong cooperativity among the tertiary interaction networks involving PK1, J2/3, PK2, and $Mg^{2+}$ binding in xrRNAs, in which PK2 plays an essential role. Although long PK2 and high $Mg^{2+}$ together can guide the proper folding of xrRNAs containing mutants at PK1 and J2/3, complete disruption of PK2 leads to misfolded nonnative conformation even at high $Mg^{2+}$ (Fig. 7). Thus, our work highlights a strategy that a delicate balance between $Mg^{2+}$ and the strength of a tertiary motif interaction (PK2) modulate the cooperativity in xrRNA to regulate its structure and function.

Our results provide mechanistic insights into in vitro Xrn1 resistance and cellular sfRNA formation mediated by flaviviral xrRNAs. An intriguing feature in most mosquito-borne flavivirus is the presence of duplicated xrRNAs (xrRNA1 and xrRNA2) in the 3′ UTR of RNA genomes, which are responsible for the generation of multiple sfRNAs in host cells[25]. Although the functions of flaviviral sfRNAs have not been fully elucidated, they are implicated to impact flavivirus replication, cytopathicity, and pathogenicity[51]. The duplication pattern of xrRNA has also been speculated to be linked to host adaptation and efficient transmission of flaviviruses[52]. Previous results show that J2/3 mutations at either xrRNA1 (PK2: 4 bp) or xrRNA2 (PK2: 2 bp) from ZIKV result in complete loss of Xrn1 resistance in vitro or sfRNA productions in infected mosquito cells[53,54], whereas J2/3 mutations of xrRNA1 (PK2: 5–7 bp) or xrRNA2 (PK2: 3 bp) from WNV partially compromise or fully abolish Xrn1 resistance in vitro, accordingly, produce low levels of sfRNA1 and essentially no sfRNA2 during infection[27]. These earlier observations

can be explained by our findings that xrRNAs with long PK2 (>5 bp) can attenuate the destructive effects of J2/3 mutations to Xrn1 resistance of xrRNAs with short PK2. Our findings suggest xrRNAs with long PK2 are more efficient in halting Xrn1, thus could lead to more sfRNA production. Consistent with this, in human cells infected with Kunjin strain of WNV which PK2 length of xrRNA1 (5–7 bp) is longer than that of xrRNA2 (3 bp), sfRNA1 produced by xrRNA1 is the most populated[26,27]. However, for DENV2 which PK2 length of xrRNA1 (2 bp) is shorter than that of xrRNA2 (4 bp), sfRNA1 is still the most populated, both in infected human and mosquito cells[55]. Thus, how the PK2 length of an individual xrRNA correlates with the pattern of cellular sfRNA production and the fitness, the pathogenicity of flaviviruses requires further exploration[26].

Our findings provide implications for potential biomaterial and biomedical applications of flaviviral xrRNAs. Because of its unique ring-like structure and mechanical properties, xrRNAs are expected to have wide applications in RNA-based nanotechnology and synthetic biology. Previous studies using single-molecule nanopore technique or molecular dynamic simulations have demonstrated the mechanical anisotropy of ZIKV-xrRNA1 which is highly dependent on $Mg^{2+}$ and tertiary interactions including two pseudoknots (PK1, PK2)[22,56]. Our results support the idea that xrRNAs with long PK2 are more stable and less-dependent on $Mg^{2+}$, implying that xrRNAs with long PK2 could be utilized as a key modular component to construct RNA-based biomaterial with extreme mechanical anisotropy. Beyond that, xrRNAs with long PK2 which are less affected by $Mg^{2+}$ should be more suitable for monitoring mRNA degradation pathways under different experimental conditions and in various biological systems[21].

## Methods

**RNA sample preparation.** The wild-type flaviviral xrRNAs and their mutant constructs were generated as follows. The plasmid encoding an upstream T7 promoter and a flaviviral xrRNA sequence was synthesized and sequenced by Wuxi Qinglan Biotechnology Inc, Wuxi, China. All the mutants were generated using Transgen's Fast Mutagenesis System. The double-stranded DNA fragment templates for in vitro RNA production were generated by PCR using a common upstream forward primer targeted the plasmids and a downstream reverse primer specific to the respective plasmids. The RNAs were in vitro transcribed using T7 RNA polymerase and purified by preparative, non-denaturing polyacrylamide gel electrophoresis, the target RNA bands were cut and passively eluted from gel slices into buffer containing 0.3 M NaOAc and 1 mM EDTA, pH 5.2 overnight at 4 °C. The RNAs were further passed through the size-exclusion chromatography column to the final buffer condition for SAXS and Xrn1 resistance experiments. The sequences for all the constructs and the primers used in this study are listed in Supplementary Tables S1 and S2, respectively.

**Site-specific internal fluorescent labeling of RNA.** The Sulfo-cyanine5 azide was purchased from Lumiprobe Cooperation. The deoxyribonucleotide phosphoramidites (dTPT3-CEP and dNaM-CEP, for DNA primer synthesis), the triphosphorylated deoxyribonucleotides (dTPT3TP and dNaMTP, for PCR) and ribonucleotides (rNaMTP, rTPT3TP, and rTPT3$^{CO}$TP, for transcription), were custom synthesized as described[35]. The procedures for site-specific internal fluorophore labeling of xrRNAs using the TPT3-NaM UBP system are similar as reported[35]. In brief, reverse primers containing unnatural nucleotides were synthesized and used to introduce the dNaMTP and dTPT3TP into the DNA template by overlap extension PCR. The PCR products containing dNaM and dTPT3 at specific sites as templates and rNTP mix supplemented with rTPT3$^{CO}$TP were used for in vitro transcription. The target RNA transcripts were purified with preparative polyacrylamide gel electrophoresis as described above. The purified xrRNAs products modified with rTPT3$^{CO}$ at specific sites were precipitated with cold ethanol (2.5 volumes) in the presence of NaOAc (0.3 M) at −80 °C for at least 0.5 h. After centrifuging at 4 °C, the ethanol was removed and the pellet was washed with ethanol (75%) three times, then the pellet was dried for 10 min. Finally, the product was resuspended in diethylpyrocarbonate-treated water and subjected to fluorescent labeling. RNAs (0.3 mM, 8 μL) were mixed with 10 μL triethylammonium acetate buffer (pH 7.0, 1 M), 25 μL DMSO, 3 μL Sulfo-cyanine5 azide (20 mM in DMSO), 1 μL of sodium ascorbate (50 mM) and 3 μL of Copper (II)-TBTA (10 mM in 55% DMSO). The resulting mixture was incubated at room temperature for 5 h. Then the labeled xrRNAs were precipitated by ethanol as described above and resuspended in buffer containing 50 mM HEPES (pH 7.5), 100 mM KCl. The labeling efficiency was calculated by measuring the absorption of

RNAs and Cy5 at 260 nm and 650 nm, respectively. The overall labeling efficiency for RNA constructs used in this study was ~80–90%.

**SAXS experiment.** SAXS measurements were carried out at room temperature at the beamline 12 ID-B of the Advanced Photon Source, Argonne National Laboratory (Supplementary Fig. 2) or the beamline BL19U2 of the National Center for Protein Science Shanghai (NCPSS) and Shanghai Synchrotron Radiation Facility (Supplementary Fig. 5). The scattered X-ray photons were recorded with a PILATUS 2 M detector (Dectris) at 12 ID-B and a PILATUS 100 k detector (Dectris) at BL19U2. The setups were adjusted to achieve scattering $q$ values of $0.005 < q < 0.89\ \text{Å}^{-1}$ (12 ID-B) or $0.009 < q < 0.415\ \text{Å}^{-1}$ (BL19U2), where $q = (4\pi/\lambda)\cdot\sin(\theta)$, and $2\theta$ is the scattering angle. Thirty two-dimensional images were recorded and reduced for each buffer or sample and no radiation damage was observed. Scattering profiles of the RNAs were calculated by subtracting the background contributed by buffer from the sample buffer profile using the program PRIMUS3.2 following standard procedures[57]. Guinier analysis was performed to calculate the forward scattering intensity $I(0)$ and the radius of gyration ($R_g$), which were also estimated from the scattering profile with a broader $q$ range of $0.006–0.30\ \text{Å}^{-1}$ using the indirect Fourier transform method implemented in the program GNOM4.6[58], along with the PDDF, $p(r)$, and the maximum dimension of the protein, $D_{max}$. Low-resolution 3D shape envelopes were ab initio reconstructed using the scattering data within $q$ range of $0.006−0.30\ \text{Å}^{-1}$ with the program DAMMIN[59], which generates models represented by an ensemble of densely packed beads.

**Xrn1 decay assay monitored by the fluorescence of MG.** The Xrn1 decay assay was performed by following the protocol developed in previous studies with minor improvement[27]. Recombinant Xrn1 from *Kluyveromyces lactis* and RppH from *Bdellovibrio bacteriovorus* was expressed in *E. coli* and purified by $Ni^{2+}$-NTA affinity and size-exclusion chromatography. In all, 90 μL reactions of 5 μM RNAs were refolded in 20 mM Tris (pH 7.5) and 100 mM KCl supplemented with different $Mg^{2+}$ concentrations. In all, 5 μL RppH (>3.5 mg/mL) and two equivalents MG were added to the reaction mixture. Then 95 μL aliquots were divided into separate wells of a black, flat bottomed 96-well view plate. Fluorescence was monitored using the Varioskan Flash plate reader. The excitation and emission wavelength were set as 630 nm and 660 nm, respectively. Fluorescence was measured for 5 min to ensure the binding between RNA and MG before adding the Xrn1, then 5 μL Xrn1 (>3.5 mg/mL) was added to the well and the reaction was monitored for the next 60 min at 37 °C. After the experiment finished, 5 μL of the mix were removed and subjected to 8% TBE-Urea denaturing gel. The fluorescence traces were normalized to controls without Xrn1 treatment based on the equation: $F = (F_{exp,t}/F_{exp,t=0})/(F_{con,t}/F_{con,t=0})$, $F_{exp}$ and $F_{con}$ refer to the fluorescence of the experimental group (+Xrn1) and control group in the absence of Xrn1 (-Xrn1), respectively. Then the normalized fluorescence trace for each RNA construct was fitted with a single exponential curve to derive the decay rate.

**Single-molecule FRET experiments.** For single-molecule experiments, 450 nM xrRNA-Cy5 was annealed with 300 nM biotin-Cy3 DNA in 50 mM HEPES (pH 7.5), 100 mM KCl by incubating the mixture at 95 °C for 2 min, then fast cooling on the ice and adding $Mg^{2+}$ to the final concentration of 100 mM, and finally equilibrated at 42 °C for 20 min. Samples were diluted 1200 times in the buffer containing 50 mM HEPES (pH 7.5), 100 mM KCl with different concentrations of $Mg^{2+}$, and immobilized on slides via biotin-streptavidin interactions. Then the samples were incubated with different concentrations of $Mg^{2+}$ for 5 min on the slide before flowing the imaging buffer containing 3 mg/mL glucose, 100 μg/mL glucose oxidase, 40 μg/mL catalase, 1 mM cyclooctatetraene, 1 mM 4-nitrobenzylalcohol, and 1.5 mM 6-hydroxy-2,5,7,8-tetramethyl-chromane-2-carboxylic acid (Trolox). smFRET experiments were conducted at 25 °C by using a home-built objective-type TIRF microscope. The time resolution of each movie was 25 ms/frame.

**Single-molecule FRET data analysis.** smFRET data were analyzed by the custom-made software program. Single-molecule movies were collected by Cell Vision software (Beijing Coolight Technology) and then analyzed by a custom-made software program developed as an ImageJ 1.43 u plugin (http://rsb.info.nih.gov/ij). Fluorescence spots were fitted by a 2D Gaussian function within a 9-pixel by 9-pixel area, matching the donor and acceptor spots using a variant of the Hough transform[38]. The background-subtracted total volume of the 2D Gaussian peak was used as raw fluorescence intensity $I$. FRET trajectories containing donor and acceptor and displaying anticorrelation behaviors were picked and analyzed. Histograms of FRET efficiency for xrRNAs at different concentrations of $Mg^{2+}$ were built by using >200 molecules. The histograms were normalized by the total number of FRET data points and the frequencies were labeled as "probability" in the y axis.

smFRET traces were further analyzed by the Hidden Markov Model-based software to extract the kinetics information[37]. Three FRET states from low- to high- FRET values were identified as $L_{state}$, $I_{state}$, and $H_{state}$ states. FRET populations for three states derived from HMM analysis were fitted to Gaussian distributions, the relative fraction of each FRET state was calculated as the ratio of

each FRET state to the total population. Transition rates between each other were estimated through the following equations,

$$k_{\mathrm{L-I}} = \frac{1}{\tau_{\mathrm{L}}} \frac{n_{\mathrm{L-I}}}{n_{\mathrm{L-I}} + n_{\mathrm{L-H}}} \qquad (1)$$

$$k_{\mathrm{L-H}} = \frac{1}{\tau_{\mathrm{L}}} \frac{n_{\mathrm{L-H}}}{n_{\mathrm{L-I}} + n_{\mathrm{L-H}}} \qquad (2)$$

$$k_{\mathrm{I-L}} = \frac{1}{\tau_{\mathrm{I}}} \frac{n_{\mathrm{I-L}}}{n_{\mathrm{I-L}} + n_{\mathrm{I-H}}} \qquad (3)$$

$$k_{\mathrm{I-H}} = \frac{1}{\tau_{\mathrm{I}}} \frac{n_{\mathrm{I-H}}}{n_{\mathrm{I-L}} + n_{\mathrm{I-H}}} \qquad (4)$$

$$k_{\mathrm{H-L}} = \frac{1}{\tau_{\mathrm{H}}} \frac{n_{\mathrm{H-L}}}{n_{\mathrm{H-L}} + n_{\mathrm{H-I}}} \qquad (5)$$

$$k_{\mathrm{H-I}} = \frac{1}{\tau_{\mathrm{H}}} \frac{n_{\mathrm{H-I}}}{n_{\mathrm{H-L}} + n_{\mathrm{H-I}}} \qquad (6)$$

in which $n_{\mathrm{L-I}}$, $n_{\mathrm{L-H}}$, $n_{\mathrm{I-L}}$, $n_{\mathrm{I-H}}$, $n_{\mathrm{H-L}}$, and $n_{\mathrm{H-I}}$ were the number of transition events captured between two specified states; $\tau_{\mathrm{L}}$, $\tau_{\mathrm{I}}$, and $\tau_{\mathrm{H}}$ were lifetimes of the low, intermediate, and high conformational states estimated by the exponential fitting of their dwell time distributions. TDPs, which depict the fraction of counts transition from a specific initial FRET state to a final FRET state among total FRET counts, were constructed from the idealized traces determined by the Hidden Markov Model[60]. And the TDPs can show the frequency of transitions between specific FRET states. Relative free energies were calculated via

$$\triangle G_b - \triangle G_a = -k_B T \bullet \ln\left(\frac{k_{a \to b}}{k_{b \to a}}\right) \qquad (7)$$

in which $k_B$ is the Boltzmann constant, $T$ is the temperature, $k_{a \to b}$ and $k_{b \to a}$ are the transition rates from the state a to b and from state b to a, respectively. $L_{\mathrm{state}}$ was set as the ground state ($\triangle G_{Low} = 0$). For illustration, the energy barrier from the state a to b was shown as $-k_B T \cdot \ln(k_{a \to b}) + 1.8 k_B T$.

**Reporting summary**. Further information on research design is available in the Nature Research Reporting Summary linked to this article.

## Data availability
The data supporting the findings of this study are available from the corresponding authors upon reasonable request. Source data are provided with this paper.

## Code availability
Custom codes used to analyze the smFRET data are provided with this paper (Supplementary Data 1).

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

## Acknowledgements

We thank the staff at beamline 12-ID-B, Advanced Photon Source, Argonne National Laboratory, and beamline BL19U2, Shanghai Synchrotron Radiation Facility, Shanghai, China for assistance during SAXS data collection. This work was supported by grants from the National Natural Science Foundation of China (nos. U1832215, 21922704, 21877069, and 22061160466), the Beijing Advanced Innovation Center for Structural Biology, the Tsinghua-Peking Joint Center for Life Sciences, to X.F. and C.C., the Beijing Frontier Research Center for Biological Structure to C.C.

## Author contributions

X.F. conceived and designed the project. C.C. and X.F. supervised the single-molecule FRET experiments. X.N., R.S. and Y.Y. performed the single-molecule FRET experiments and analyzed the data. X.N. and Z.C. performed an activity assay and prepared all the RNA samples, Z.C. and X.N. analyzed the SAXS experiments. X.N., C.C. and X.F. wrote the manuscript with input from the other authors.

## Competing interests

The authors declare no competing interests.
