## [Peer Review File · Nature Communications]

Pseudoknot length modulates the folding, conformational dynamics and robustness of Xrn1 resistance of flaviviral xrRNAsREVIEWER COMMENTS

Reviewer #1 (Remarks to the Author):

Manuscript:

Pseudoknot length modulates the folding, conformational dynamics, and robustness of Xrn1 resistance of flaviviral xrRNAs

Review:

In this manuscript by Niu et al., the authors examine the relationship between pseudoknot structure and Xrn1 resistance in 11 naturally occurring flavivirus pseudoknots. The authors use several techniques to probe pseudoknot structure and dynamics, including bulk Xrn1 cleavage assays, small-angle x-ray scattering (SAXS), and single-molecule fluorescence resonance energy transfer (smFRET). The authors conclude that increasing the length of the PK2 region of the pseudoknot results in increased stability of the pseudoknot structure, which enhances Xrn1 resistance. They further use kinetic modeling to demonstrate that less Mg²⁺ is required to stabilize pseudoknots with longer PK2. Lastly, they alter key pseudoknot tertiary interactions outside of the PK2 region and note that pseudoknots with longer PK2 regions are more capable of keeping their pseudoknot structures in the presence of destabilizing mutations. The authors present their data with well-organized figures which, for the most part, are both visually appealing and clear.

While the data presented are convincing, there are a few controls missing, which limits enthusiasm for the publication of this manuscript in Nature Communications. One major concern is that while the authors do observe a correlation between PK2 length and pseudoknot structure/dynamics/Xrn1 resistance, they never manipulate the length of any PK2 to test if altering PK2 length directly results in a change in RNA structure/dynamics or sensitivity to Xrn1, or introduce compensatory mutations to ensure that it is not sequence context that matters. Because the PK2s are all found in different structural contexts, it is not clear if there are other regions that influence their activity.

In addition to concerns about controls, there are also a few instances in the manuscript where the depth of quantitative analysis needs to be increased. The manuscript would be enhanced if the authors provided more description into the quality of their fitting with the HMM, by way of transition density plots (TDPs) or other analysis models. Furthermore, in regard to tying the Xrn1 resistance activity to pseudoknot structure, the agreement of the Xrn1 decay assays with structural and smFRET data could be more rigorously analyzed.

In terms of overall impact, a deeper discussion of the biological context and impact of PK2 length is lacking. Additionally, the quality of grammar and language in the manuscript needs some improvement not to be a major distraction. Grammatical errors make it difficult to understand the information that the authors are attempting to convey in several instances. Finally, the biological context of PK2 was not well discussed, or explored in light of the Xrn1 resistance data. How does increased Xrn1 resistance impact the intracellular fitness of the viruses studied?

With additional control experiments and major revisions to this manuscript, it is possible that it could be brought up to the standards expected for publication in Nature Communications. The topic will be of interest to the field, and the experiments are elegantly designed, especially the smFRET assay.

Listed below are major and minor concerns which should be addressed by the authors.

Major Concerns:

1. While the authors test several different flavivirus pseudoknots, they do not directly test in any case

whether altering PK2 length within the same sequence context changes the structure/dynamics of the pseudoknot or its sensitivity to Xrn1 and Mg²⁺. It is critical that the authors vary the lengths of PK2 either in isolation (with just the 5' hairpin region to the end of the PK2 sequence) or in the full natural constructs to take this information from an interesting correlation to a tested hypothesis, including the introduction of compensatory mutations to show that it is truly helix length that matters most. This could be addressed by taking the three pseudoknot structures they use for smFRET experiments (DENV2, ZIKV, and WNV) and simply increasing or decreasing the PK2 region in length.

In addition, the PK2-L3 mutation is not properly explained in the text. Does it still allow to form the PK2? Did the authors try a mutation where the PK2 is not formed at all? That would help make it clear that PK2 is required to maintain the folded structure and the Xrn1 resistance activity.

2. From Fig. 1c (bottom panel): It looks like the RNA with only a 2-bp PK2 is more folded than that with a 3- or 4-bp PK2. The authors should discuss this observation – not consistent with their preferred interpretation – in the Results section and a possible reason behind it.

3. The Kratky plots and associated R_g values as a function of Mg²⁺ concentration should be utilized to derive structural models that evaluate just how folded the different xrRNAs are. For example, structural models such as those derived by ATTRACT-SAXS (PMID 27427479) or similar programs would allow the authors to make model-independent determinations of the unfolded or partially folded conformations, and derive mechanistic insights into what distinguishes them from the fully folded pseudoknot.

4. The site-specific labeling strategy is very interesting. However, it is not sufficiently clear how it was performed. A detailed description of the preparation of these labeled RNAs would be helpful for readers to understand and enable a broader use of this technique in the future. In addition, did the authors actually test site-specificity and labeling efficiency? If so, what are they?

5. It is unclear from the Results section how the SAXS data are correlated with the smFRET data. Although the authors mention their consistent observation of the folded RNA and high-FRET state in SAXS and smFRET, respectively, how they interpret the partially folded and unfolded data in light of the SAXS data is unclear. Again, SAXS data at low Mg²⁺ concentration might help to clarify; see also point 3.

6. There are several areas where an increase in quantitative analysis of data already shown would greatly enhance the manuscript. For example:

a. The authors do not have any metric to describe how representative the smFRET traces are of their datasets. Use of transition density plots (TDPs), or perhaps better transition occupancy density plots (TODPs) and other analytical strategies, would provide this information. A TODP could further show the frequency of transitions between specific FRET states, as well as any unchanging states, which could clarify the mechanism of folding in more detail. Further, the authors do not describe how many molecules were used to construct each of the histograms in Figures 3D and 6D. This should be made clear for each plot.

b. The authors describe rate constants for Xrn1 resistance in their fluorescence experiments, but they do not tie those observations to the rate constants calculated. For example, it would be interesting to note what the probability of decay was based on the “openness” of the structure, and how well that agrees with the actual observed decay.

7. Supplementary Figure 3 should be included as a main figure, since the impact of PK2 on Xrn1 resistance is one of the major findings in this manuscript. It also appears that the authors did not conduct replicates of these experiments. Although the data show a clear dependence on [Mg²⁺], especially for the “long” PK2s, this set of experiments should be repeated at least twice to derive error estimates.

8. For Supplementary Figure 3 panels h-j, why did the authors only go down to 0.1 mM and up to 5 mM Mg²⁺ instead of from 0.001 mM to 20 mM as for panels k-m? Because of the smaller range, the “medium” PK2 lengths appear more “resistant” to Xrn1 cleavage activity than the “long” PK2 lengths. For this reason, the low concentrations seem especially important to include. All 9 [Mg²⁺] are not necessary, but at least the 0.001 mM Mg²⁺ should be done.

9. It would be important for the authors to discuss their findings in the light of biological relevance of flavivirus Xrn1 resistance. For instance, does increased Xrn1 resistance result in less favorable outcomes in disease? Does it result in shorter viral cycles in cells? Why would some viruses need a shorter PK2? Some commentary would increase the breadth of interest for the manuscript. In addition, all prior studies as to the relevance of the PK2 length (such as current references 56 and 57) should be presented in the Introduction as motivation for the current studies, rather than only as an afterthought at the very end.

10. In the results section authors noted that “.....Although Hstate can directly transit to Lstate under almost all conditions, its reverse reaction, direct transition from Lstate to Hstate, is extremely slow and rarely seen, even at high Mg²⁺ when Hstate is the most stable state. These phenomena strongly support that the folding of xrRNAs follows a definite sequential pathway corresponding to transitions from Lstate to Istate, then to Hstate.....” –

Is this totally correct?

- For example, in WNV-xrRNA 1, Fig. 3c, it seems that the folded to unfolded transition always happens through the partially folded state. This should be noted in the Results section with a possible explanation.

- From the traces shown in Fig. 3a-c, the sequential pathway from L to I to H seems not always followed. This part of the results section needs major correction to clarify the folding mechanism in more detailed manner. Again, TDP/TODP plots would greatly help lend statistical significance to the stated observations. In addition, one can estimate (from the rate constants) how likely it is for I to be skipped and a direct L-to-H transition to instead be observed – is this expectation met?

11. In the Discussion, there are instances where the same result is discussed multiple times. Instead, a considerably more thorough contextualization with respect to the known data on xrRNA should be presented.

12. Figure 1 should indicate where the mutations were later made to guide the reader.

13. Figure 2 and 3 could be combined because they refer to the same experiment.

14. Figures 3 and 6: Often with smFRET trace fitting, a hidden Markov model is overlaid on top of the traces to demonstrate how well the models describe the data. Figures 3A and 6A would be improved with the incorporation of fits displayed over the traces (include example) as that would make the assignments of the three purported states much clearer.

15. Did the authors consider any “static” traces in the derivation of their rate constants? If molecules do not change state over the relatively short observation windows used here, then they might transition during the unobserved next instance, which would skew the reported rate constants. In addition, can the authors’ quantification of the rate and equilibrium constants be related back to the kinetics and thermodynamics of Xrn1’s degradation action? Xrn1 is a motor protein that converts phosphodiester hydrolysis into a force for forward movement, counteracted by the force resistance of the pseudoknot; alternatively, degradation could be kinetically controlled by the relative rates of opening/closing of the pseudoknot and that of Xrn1’s advance on the RNA. The authors should have this information, and by such an additional, more sophisticated analysis could substantially strengthen the biological significance and impact of their work.

16. Their transition state analysis tells the authors that the transition state is more similar to the I

state than the H state – they should be more explicit about broader conclusions such as this from their transition state theory in Figures 4 and 6.

17. For Figure 6, the authors should discuss what the significance is that the I state shifts, in some cases quite dramatically, as is the case for the PK2-L3 mutant of WNV.

Minor Concerns:

1. There are many grammatical issues throughout the manuscript, which at times distract from the message, especially in the Introduction and Discussion sections. These issues must be addressed prior to resubmission. Some examples are listed below:

a. Line 64: “Despite of the divergence...” should be “Despite the divergence...”.

b. Line 103: “...and mutations that disrupt key tertiary interactions including PK1, PK2 and J2/3 using single-molecule Fluorescence Resonance Energy Transfer (smFRET).” is confusing.

c. Line 144: “...fluorescence-based Xrn1 decay assay (Supplementary Fig. 3a) was performed...” should be “...a fluorescence-based Xrn1 decay assay (Supplementary Fig. 3a) was performed...” or “...fluorescence-based Xrn1 decay assays (Supplementary Fig. 3a) were performed...”

d. Line 177: “intent” should be “intend”.

e. Line 440: “An ensemble description of RNA structure has been increasingly applied to clarify RNA folding, conformational dynamics and understand RNA function.”

f. Line 446: “Recent biophysical studies showed that ZIKV-xrRNA1 form alternative” should be “forms alternative”.

g. These are among many other examples throughout the manuscript.

2. On page 4, lines 73 and 75, it is unclear what the authors mean by “mechanical anisotropy”.

3. On page 5, line 91, the authors refer to the “undocked” and “docked” conformations. It is unclear at this point what that means. Could the authors expand on this statement, and point more clearly in Figure 1b as to what they mean by “undocked” and “docked” conformations?

4. For Figure 1f, the authors do not clearly state what strategy (either the gel assays or fluorescence assays) is used to produce the decay rate curves. This should be made clear in both the figure legend and the text. In addition, the positions of panels c and f should be switched for a more canonical layout.

5. In Supplementary Figure 3, it is unclear what the decay products are in the denaturing PAGE gels. Are these the end products for each [Mg²⁺] at 60 minutes? This should be noted in the legend.

6. In Figure 4, it might be helpful to the audience if the PK2 length was mentioned in the legend or titles of the plots, to help remind readers that these xrRNAs represent “short”, “medium”, and “long” PK2s.

7. The colors used in Figure 4c-f for 1 mM and 100 mM Mg²⁺ are very similar, and hard to distinguish. The authors may want to consider adopting the color scheme used in Figure S7, which provides much more distinction.

8. The normalization strategy (i.e., the equation used) for the Xrn1 decay kinetics assay should be provided or at least described in words.

9. In the Methods section, it looks like “°C” is not recognized as a character for the PDF version of the manuscript. It appears as a square. This should be amended in future versions.

10. The kbT values should (also) be given as more canonical kcal/mol values.

Reviewer #2 (Remarks to the Author):

Exonuclease-resistant RNA pseudoknots (xrRNAs) have attracted considerable interest in recent years because of their importance in flaviviruses where they first discovered, and their usefulness in RNA technologies. This manuscript very thoroughly describes how the length of pseudoknot 2 (PK2) affects the stability of xrRNA tertiary structure and the ability to resist degradation by Xrn1 exonuclease. This follows up the NComm 2020 paper by the same authors on the mechanical stability of xrRNA (using a nanopore).

The strength of this study is its thoroughness – the authors begin by comparing 11 (!) xrRNA motifs from different viruses, and then investigate three examples in more depth. These examples have different numbers of base pairs in PK2. It is commendable that the authors looked at so many different natural xrRNA sequences in order to accurately assess the importance of PK2 length. Another strength of this study is that the authors studied the folding of the xrRNAs with exonuclease assays and with biophysical methods, SAXS and smFRET, that provide complementary information about the folding equilibrium and kinetics. Finally, they use targeted mutations to evaluate the cooperativity between PK2 and other elements of xrRNA tertiary structure.

The final conclusion is not surprising – a longer PK2 pairing correlates with a more stable fold and better exonuclease resistance. That PK2 is important for exonuclease resistance was shown in the earliest papers on this motif and the authors showed in their 2020 paper that PK2 is particularly important for mechanical stability. The authors don't really connect their findings to the biological roles of these motifs in different viruses, although presumably other authors could do that. The main value of this study may be for xrRNA engineering. Perhaps the authors could comment on that a bit more.

Although a few points listed below require clarification, the experiments were well-done, and the manuscript is written clearly.

Specific comments:

1. The SAXS experiments nicely show that all of the natural xrRNAs fold to a compact structure in 5 mM MgCl₂. The authors emphasize that the ones with the longest PK2 are also able to fold in EDTA, based on R_g and D_{max}. However, the reduced Kratky plots in Fig. 5c show that even the most stable sequence is very clearly less compact in EDTA than it is in 5 mM MgCl₂. This is presumably due to transient unfolding of the RNA tertiary structure but could also arise from non-native structure as the authors allude to later. I think the authors should revise their text on lines 139-141 to acknowledge this possibility. It is possible that none of the xrRNAs achieve the true native structure without some Mg²⁺, although they can get close if PK2 is stable enough. Evidence for this is also in Fig. 5ab.

2. The authors use a non-standard base to introduce the Cy3 dye next to PK2 for smFRET studies. They state that this modification has only a minor effect on the stability of the RNA tertiary structure, but the data showing this in Fig. S4 are very brief. What length of Xrn1 treatment and Mg²⁺ concentration was used to test the stabilities of the modified RNAs? How sensitive was this test, and is it a fair comparison without a leader that can be grasped by Xrn1?

3. The potential for interference by the fluorophore is relevant to interpreting the smFRET experiments. In general, these data are very lovely, but as the authors note on lines 235-236 and 258, even the most stable xrRNA studied continues to fluctuate between the folded and unfolded states in 100 mM MgCl₂. Conditions that yield maximum Xrn1 resistance correlate with as little as 45% occupancy of the folded state. These observations beg the question of how fluctuating xrRNAs can resist the exonuclease – is there some minimum dwell time in the unfolded state required to let Xrn1 by? Or, is the fluorophore making the structures more dynamic, leading to this apparent discrepancy? If the latter were true, I don't think it would undermine the point of this paper (a

comparison of PK2 lengths), but it would be important to measure any perturbation to arrive at the proper interpretation. (NB: it is possible for the fluorophore to perturb the folding and unfolding rate constants without a large change in the folding equilibrium.)

4. The FRET efficiencies are fit to three states in Fig. 3, but the data suggest there may exist one or more "hidden" intermediate states that the authors should consider (or at least discuss). First, the wt xrRNAs seem to form folded (high FRET) states with different lifetimes. Sometimes this shows up as fluctuations in the Cy5 channel, and sometimes as FRET changes that are short lived (eg ZIKV in 1 mM MgCl₂). Second, the average FRET values of the L, I and H populations shift with MgCl₂ concentration in the histograms (Fig. 3d-f), which should not be the case if the system is merely sampling three states. This possibility becomes more apparent for the mutants in Fig. 6. I recommend that the authors look more carefully at the data and consider whether a more complicated folding model would help them better explain the effects of the mutations and the link between folding dynamics and exonuclease resistance.

5. In Figure 6g, the authors state that the mutant xrRNAs are misfolding, yet this is not shown in the free energy landscape. This diagram should be modified to indicate a fourth conformational state in keeping with their interpretation.

6. What transmission frequency (barrier free rate constant) was assumed for the folding reactions when building the free energy landscapes (line 642), and how was this chosen?

7. The authors should provide more information about the numbers of molecules used for each smFRET experiment and the numbers of observations, as well as the error analysis. Perhaps the number of molecules can be included in the supplemental tables.

Reviewer #3 (Remarks to the Author):

The manuscript of Xiaolin Niu and co-authors reports the study on thermodynamic stability and exonuclease resistance of flaviviral xrRNAs, the conserved XRN-1-resistant RNA elements in 3'UTRs of flaviviruses responsible for biogenesis of viral noncoding RNA (sfRNA). The exonuclease resistance of xrRNAs is achieved due to the unique ring-like structure of these elements, which is determined by canonical and noncanonical base pairing interactions within the structure. The most critical among them are two pseudoknots (small pseudoknot PK1 and typically large pseudoknot PK2) and interactions of unpaired C nucleotide (J2/3). By using the range of biochemical and biophysical techniques the authors demonstrated that formation of PKs are Mg²⁺-dependent, however the concentration of magnesium required for PK formation depends on length of PK2 with longer PKs requiring less Mg²⁺. Authors also demonstrated that xrRNAs exist in three conformational states – the stable XRN1-resistant fold and two transitional folds that are not XRN1 resistant. xrRNAs spend half of their time in XRN1-resistant state and present only transiently in nonrestraint states. Authors demonstrated that higher Mg²⁺ concentration and longer PK2 stabilise XRN-1 resistant state. As the result xrRNAs with longer PK2 (7nt) can tolerate mutations in PK1 and J2/3. Although the requirement of Mg²⁺ for XRN1 resistance of xrRNAs is known (PMID: 24744377), the rest of the results represent novel findings that improve our mechanistic understanding of XRN-1 resistance.

However, the biological implications of these purely in vitro findings are not investigated or even discussed in the manuscript. These include:

1. What are the implications of Mg²⁺ concentration on different conformational states of xrRNAs in the physiologically relevant Mg²⁺ concentrations in virus-infected cells (0.5-1mM, e.g. PMID: 29920012)
2. What are the roles of different conformational states of xrRNA in virus replication and virus-host interactions?

3. Is there any correlation between PK2 length and pathogenicity or epidemiological fitness of the viruses?
4. Is there an evolutionary trend in increasing PK2 length?

Response to reviewer 1

Referee #1:

In this manuscript by Niu et al., the authors examine the relationship between pseudoknot structure and Xrn1 resistance in 11 naturally occurring flavivirus pseudoknots. The authors use several techniques to probe pseudoknot structure and dynamics, including bulk Xrn1 cleavage assays, small-angle x-ray scattering (SAXS), and single-molecule fluorescence resonance energy transfer (smFRET). The authors conclude that increasing the length of the PK2 region of the pseudoknot results in increased stability of the pseudoknot structure, which enhances Xrn1 resistance. They further use kinetic modeling to demonstrate that less Mg^{2+} is required to stabilize pseudoknots with longer PK2. Lastly, they alter key pseudoknot tertiary interactions outside of the PK2 region and note that pseudoknots with longer PK2 regions are more capable of keeping their pseudoknot structures in the presence of destabilizing mutations. The authors present their data with well-organized figures which, for the most part, are both visually appealing and clear.

While the data presented are convincing, there are a few controls missing, which limits enthusiasm for the publication of this manuscript in Nature Communications. One major concern is that while the authors do observe a correlation between PK2 length and pseudoknot structure/dynamics/Xrn1 resistance, they never manipulate the length of any PK2 to test if altering PK2 length directly results in a change in RNA structure/dynamics or sensitivity to Xrn1, or introduce compensatory mutations to ensure that it is not sequence context that matters. Because the PK2s are all found in different structural contexts, it is not clear if there are other regions that influence their activity.

In addition to concerns about controls, there are also a few instances in the manuscript where the depth of quantitative analysis needs to be increased. The manuscript would be enhanced if the authors provided more description into the quality of their fitting

with the HMM, by way of transition density plots (TDPs) or other analysis models. Furthermore, in regard to tying the Xrn1 resistance activity to pseudoknot structure, the agreement of the Xrn1 decay assays with structural and smFRET data could be more rigorously analyzed.

In terms of overall impact, a deeper discussion of the biological context and impact of PK2 length is lacking. Additionally, the quality of grammar and language in the manuscript needs some improvement not to be a major distraction. Grammatical errors make it difficult to understand the information that the authors are attempting to convey in several instances. Finally, the biological context of PK2 was not well discussed, or explored in light of the Xrn1 resistance data. How does increased Xrn1 resistance impact the intracellular fitness of the viruses studied?

With additional control experiments and major revisions to this manuscript, it is possible that it could be brought up to the standards expected for publication in Nature Communications. The topic will be of interest to the field, and the experiments are elegantly designed, especially the smFRET assay.

Response #1: We appreciate the reviewer's positive comments and constructive suggestions on our work and have addressed all the major and minor concerns point-by-point as below.

Listed below are major and minor concerns which should be addressed by the authors.

Major Concerns:

1. While the authors test several different flavivirus pseudoknots, they do not directly test in any case whether altering PK2 length within the same sequence context changes the structure/dynamics of the pseudoknot or its sensitivity to Xrn1 and Mg²⁺. It is critical that the authors vary the lengths of PK2 either in isolation (with just the 5' hairpin region to the end of the PK2 sequence) or in the full natural constructs to take

this information from an interesting correlation to a tested hypothesis, including the introduction of compensatory mutations to show that it is truly helix length that matters most. This could be addressed by taking the three pseudoknot structures they use for smFRET experiments (DENV2, ZIKV, and WNV) and simply increasing or decreasing the PK2 region in length.

Response #2: We thank the reviewer's constructive comments. As suggested, PK2 mutants of both ZIKV-xrRNA1 (ZIKV-PK2mut1, ZIKV-PK2mut2) and WNV-xrRNA1 (WNV-PK2mut1, WNV-PK2mut2) were generated, in which PK2 lengths were shortened from 4 bp to 3 bp, 2bp, and from 6 bp to 4 bp, 2 bp, respectively (Supplementary **Fig. 5a**). The Mg²⁺-dependence of their folding and Xrn1 resistance activity were studied by SAXS and Xrn1 decay kinetics assay (Supplementary **Fig. 5b-e**). These results nicely support a correlation between PK2 length and Mg²⁺-dependence of xrRNAs' folding and robustness of Xrn1 resistance. It would be challenging to increase the PK2 length by introducing new base pairs in DENV2-xrRNA1 (PK2: 2 bp) due to the potential topological restraint of the native structure, so we limited the analysis to ZIKV-xrRNA1 and WNV-xrRNA1 variants. The discussion of the results and a new supplementary **Fig. 5** were added to the revised manuscript.

In addition, the PK2-L3 mutation is not properly explained in the text. Does it still allow to form the PK2? Did the authors try a mutation where the PK2 is not formed at all? That would help make it clear that PK2 is required to maintain the folded structure and the Xrn1 resistance activity.

Response #3: To present the results about PK2-L3 mutants for more clarity, we revised the text and added the secondary structures of PK2-L3 mutants in supplementary **Fig. 8** (supplementary **Fig. 6** in old version). Previous studies on PK2 mutants of xrRNAs from ZIKV and WNV have tested the significance of PK2 in conveying *in vitro* Xrn1 resistance and cellular sRNA formation (PMID: 20719943, 27934765). Among these PK2 mutants, the PK2s in WNV-xrRNA1 were only partially disrupted (PMID: 20719943), and the PK2s in ZIKV-xrRNA1 were completely disrupted (PMID:

27934765), but the Mg^{2+} -dependence of their Xrn1 resistance has not been tested. By contrast, in our PK2-L3 mutants which the nucleotides in L3 loops potentially involved in PK2 formation were all mutated to the complementary sequences (same as that in S4 loop) (supplementary **Fig. 8a-c**), PK2 will not formed at all in principle. Xrn1 decay kinetics assays show that the PK2-L3 mutants of all three xrRNA1s abolish Xrn1 resistance activities at any Mg^{2+} concentrations (including 5 mM Mg^{2+}) (**Figure 5e-g**, supplementary **Fig. 8j-l**). These results are significantly different from that of the PK1 and J2/3 mutants, which sample native conformations in xrRNA with long PK2 (WNV-xrRNA1) in high Mg^{2+} (5 mM), thus suggesting an essential role of PK2 in maintaining the native folding and Xrn1 resistance activity of xrRNAs. We have revised the manuscript accordingly.

2. From Fig. 1c (bottom panel): It looks like the RNA with only a 2-bp PK2 is more folded than that with a 3- or 4-bp PK2. The authors should discuss this observation – not consistent with their preferred interpretation – in the Results section and a possible reason behind it.

Response #4: We appreciate the reviewer's comment. To gain more confidence with our data, we recently made a new batch of the 11 xrRNAs samples and repeated the SAXS measurements in 5 mM EDTA at the APS SAXS beamline. We basically reproduce the observations that xrRNAs with long PK2 (> 5 bp) folded more compact than those with short PK2 (< 5 bp) in 5 mM EDTA. But it's very likely that there is no strict pattern between PK2 length and folding for xrRNAs with short PK2 (< 5 bp) in EDTA. There could be several reasons. Firstly, besides PK2 length, the variations of sequences and other tertiary interactions of different xrRNAs may contribute to the folding, thus complicate the pattern. Actually, the pattern between folding and PK2 length becomes clear in the context of the same xrRNAs (ZIKV-xrRNA1 and WNV-xrRNA1). As suggested by the reviewer, we made PK2 mutants for ZIKV-xrRNA1 and WNV-xrRNA1 in which the PK2 lengths were decreased. SAXS data in EDTA show that the folding of these xrRNA1 variants becomes less compact when the PK2 length

is shorter (supplementary **Fig. 5**). Secondly, the predicted secondary structures and PK2 lengths by bioinformatics analysis were validated experimentally only for several of the 11 flaviviral xrRNAs (ZIKV-xrRNA1 and MVEV-xrRNA2) in this work. Therefore, we revised the text and did not claim any definite pattern for wild type xrRNAs in EDTA. Instead, we emphasize the Mg^{2+} -dependence of xrRNAs' folding (the differences between EDTA and Mg^{2+}) on PK2 length, which is more obvious across the 11 xrRNAs.

3. The Kratky plots and associated R_g values as a function of Mg^{2+} concentration should be utilized to derive structural models that evaluate just how folded the different xrRNAs are. For example, structural models such as those derived by ATTRACT-SAXS (PMID 27427479) or similar programs would allow the authors to make model-independent determinations of the unfolded or partially folded conformations, and derive mechanistic insights into what distinguishes them from the fully folded pseudoknot.

Response #5: We thank for the suggestions and studied the references and programs mentioned by reviewer. The ATTRACT-SAXS is used to derive structural model of protein-protein complex, which requires the input of two protein structure and SAXS experimental data. Thus, it is not appropriate to derive structural model of individual RNA molecule using ATTRACT-SAXS. We therefore *ab initio* reconstruct 3D shape envelopes for all the xrRNAs, both in Mg^{2+} and in EDTA, using the program DAMMIN, which models a macromolecule as an assembly of scattering beads arranged in space such that a calculated scattering curve reproduces the experimental curve (PMID: 10354416). The results were shown in Supplementary **Fig. 3**. Clearly, for xrRNAs with short PK2 (<5 bp), the global structures are more extended in EDTA than that in Mg^{2+} , for xrRNAs with long PK2 (>5 bp), their global structures become compact even in EDTA, which are close to that in Mg^{2+} . As SAXS is an ensemble technique, it probes the structural features by averaging all the conformations in solution. Currently, it's difficult to directly distinguish the unfolded, partially folded, and folded structures from

the SAXS data without complex computational modeling, which however is not the focus of this work.

4. The site-specific labeling strategy is very interesting. However, it is not sufficiently clear how it was performed. A detailed description of the preparation of these labeled RNAs would be helpful for readers to understand and enable a broader use of this technique in the future. In addition, did the authors actually test site-specificity and labeling efficiency? If so, what are they?

Response #6: We appreciate reviewer's positive comments on the site-specific labeling strategy. We have revised both the text including the **Methods** and **Figure 2**. The site-specificity has been tested by including or omitting rTPT3TP/rTPT3^{CO}TP in the rNTP mix during *in vitro* transcription, which is also reported in recent work (PMID: 32868439, 33224460). The labeling efficiency was calculated by measuring the absorption of RNAs at 260 nm and 650 nm and reported as 80-90% overall. Thus, the UBP-based fluorescence labeling strategy is site-specific and highly efficient.

5. It is unclear from the Results section how the SAXS data are correlated with the smFRET data. Although the authors mention their consistent observation of the folded RNA and high-FRET state in SAXS and smFRET, respectively, how they interpret the partially folded and unfolded data in light of the SAXS data is unclear. Again, SAXS data at low Mg²⁺ concentration might help to clarify; see also point 3.

Response #7: RNA can be described as an ensemble of interconverting conformers with unfolded, partially folded or folded conformations in solution. SAXS and smFRET have emerged as powerful techniques to explore the folding and conformational dynamics of RNAs in near physiological conditions. While SAXS is an ensemble technique, which probes the global structural features by averaging all the conformations in solution, smFRET can provide RNA structure and dynamics information at single-molecule level. As shown in our work, at low Mg²⁺ concentrations

(e.g. 5 mM EDTA or 0.001 mM Mg^{2+}), the SAXS data indicates that some xrRNAs (e.g. DENV2-xrRNA1) are conformationally extended from *ab initio* shape envelopes, or unfolded or partially folded inferred from the dimensionless Kratky plots, and smFRET data suggest that xrRNAs mainly sample low FRET or intermediate FRET states in the ensembles. Similarly, in 5 mM Mg^{2+} , SAXS data indicate that xrRNAs are well folded, smFRET data suggest the xrRNAs still sample three different conformations, though the high FRET state is mostly populated. Therefore, SAXS and smFRET are complementary techniques that provide distinct conformational properties of the system. With the aid of computational modeling, SAXS and smFRET in combination are expected to provide more complete conformational description of xrRNA ensemble.

6. There are several areas where an increase in quantitative analysis of data already shown would greatly enhance the manuscript. For example:

a. The authors do not have any metric to describe how representative the smFRET traces are of their datasets. Use of transition density plots (TDPs), or perhaps better transition occupancy density plots (TODPs) and other analytical strategies, would provide this information. A TODP could further show the frequency of transitions between specific FRET states, as well as any unchanging states, which could clarify the mechanism of folding in more detail. Further, the authors do not describe how many molecules were used to construct each of the histograms in Figure 3D and 6D. This should be made clear for each plot.

Response #8: We appreciate the reviewer's suggestions. As suggested, we have performed transition density plots (TDP) analysis on DENV-xrRNA1, ZIKV-xrRNA1 and WNV-xrRNA1 in different Mg^{2+} concentrations (**Figure 3**, Supplementary **Fig. 7d-f**), and on PK1, J2/3 and PK2-L3 mutants of DENV-xrRNA1, ZIKV-xrRNA1 and WNV-xrRNA1 at high Mg^{2+} concentrations (**Figure 6**). We have discussed the results of TDP analysis and updated figures in the revised manuscript.

b. The authors describe rate constants for Xrn1 resistance in their fluorescence experiments, but they do not tie those observations to the rate constants calculated. For example, it would be interesting to note what the probability of decay was based on the “openness” of the structure, and how well that agrees with the actual observed decay.

Response #9: We thank the reviewer for the insightful comments. The enzymatic decay rates for xrRNAs were measured as 0.005-0.34 min⁻¹ in bulk Xrn1 resistance assay, in contrast, the transition rates between different FRET states were calculated as 0.03-5.43 s⁻¹ in smFRET experiments. As these two rates were not in the same timescale and can't be compared directly, we carried out correlation analysis between the decay rates and transition rates in different Mg²⁺ concentrations.

As shown in **Figure 4g-j**, the Xrn1 decay rates of DENV-xrRNA1, ZIKV-xrRNA1 and WNV-xrRNA1 show positive correlation with the transition rates of k_{H-I} and k_{I-L} at different Mg²⁺ concentrations, but no significant correlation can be observed for k_{I-H} and k_{L-I} , indicating that the faster xrRNAs escape from the H_{state} and I_{state} , the easier for them to be degraded by Xrn1. These analyses reveal that the degradation of xrRNAs by Xrn1 is kinetically controlled by the unfolding process, defined as transitions of xrRNAs from high FRET to intermediate or low FRET states. In other words, decay of xrRNAs by Xrn1 is based on the openness of its ring-like structure. We discuss these new findings in the revised text and add new figures as **Figure 4g-k**.

7. Supplementary Figure 3 should be included as a main figure, since the impact of PK2 on Xrn1 resistance is one of the major findings in this manuscript. It also appears that the authors did not conduct replicates of these experiments. Although the data show a clear dependence on [Mg²⁺], especially for the “long” PK2s, this set of experiments should be repeated at least twice to derive error estimates.

Response #10: We agree with the reviewer that the impact of PK2 on Xrn1 resistance is one of the major findings in our work. The Xrn1 decay rates for the respective xrRNAs at different Mg²⁺ concentrations were derived from the data in Supplementary **Fig. 4** (S3 in old version). Since the Mg²⁺-dependence of decay rates for the respective

xrRNAs have been plotted in **Figure 1f**, we decide not to include current Supplementary **Fig. 4** as a main figure. For each xrRNA in Supplementary **Fig. 4**, the Xrn1 decay kinetics assay has been repeated for at least three times in different Mg^{2+} concentrations and the kinetic curves were presented as mean \pm standard error of the mean which were calculated from three independent experiments.

8. For Supplementary Figure 3 panels h-j, why did the authors only go down to 0.1 mM and up to 5 mM Mg^{2+} instead of from 0.001 mM to 20 mM as for panels k-m? Because of the smaller range, the “medium” PK2 lengths appear more “resistant” to Xrn1 cleavage activity than the “long” PK2 lengths. For this reason, the low concentrations seem especially important to include. All 9 [Mg^{2+}] are not necessary, but at least the 0.001 mM Mg^{2+} should be done.

Response #11: We thank the reviewer for the constructive suggestions. As suggested, we have conducted Xrn1 decay kinetics assay for all xrRNAs in 0.001-20 mM Mg^{2+} concentrations and updated the Supplementary **Fig. 4** and **Figure 1f**. We also revised the manuscript to discuss the results. From **Figure 1f**, clearly, xrRNAs with short PK2 (< 5 bp) requires higher Mg^{2+} (such as 0.5-1 mM) to resist the degradation by Xrn1, while xrRNAs with long PK2 (> 5 bp) exhibit significant Xrn1 resistance at even low Mg^{2+} concentrations (such as 0.1-0.2 mM).

9. It would be important for the authors to discuss their findings in the light of biological relevance of flavivirus Xrn1 resistance. For instance, does increased Xrn1 resistance result in less favorable outcomes in disease? Does it result in shorter viral cycles in cells? Why would some viruses need a shorter PK2? Some commentary would increase the breadth of interest for the manuscript. In addition, all prior studies as to the relevance of the PK2 length (such as current references 56 and 57) should be presented in the Introduction as motivation for the current studies, rather than only as an afterthought at the very end.

Response #12: The most important biological functions of xrRNA relates to its *in vitro*

Xrn1 (or other exoribonucleases) resistance and cellular sfRNA (subgenomic flaviviral RNA) formation. Our study reveals an important regulatory role of PK2 length in modulating the structure, conformational dynamics and Xrn1 resistance of flaviviral xrRNAs, providing mechanistic insights to the *in vitro* and *in vivo* functions of xrRNAs.

Inspired by the reviewer's comments, we have revised the manuscript accordingly. (1) We have included prior studies on the significance of PK2 length to xrRNAs' function in the introduction section; (2) We have discussed the biological implication of our findings to cellular sfRNA formation. Our findings imply that xrRNAs with long PK2 are more efficient in stalling Xrn1 *in vitro*, thus the increased Xrn1 resistance activity may lead to the production of more sfRNAs *in vivo*. Although the functions of flaviviral sfRNAs have not been fully elucidated, they are implicated to impact on flavivirus replication, cytopathicity and pathogenicity.

However, it's difficult to directly relate PK2 length in xrRNAs to the fitness and pathogenicity of flaviviruses. An intriguing feature in most mosquito-borne flavivirus (MBFV) is the presence of duplicated xrRNAs (xrRNA1 and xrRNA2) in the 3' UTR of RNA genomes. This duplication pattern has been linked to the generation of multiple sfRNAs in flaviviruses, adaptation to different hosts and efficient transmission, etc (PMID: 32581095, 32581095). We analyzed the PK2 lengths of xrRNA1 and xrRNA2 across different flaviviruses and found no clear pattern (**Response Fig. 1**). In DENV1 and DENV2, the PK2 lengths in xrRNA1s are shorter than that in xrRNA2s, however, the PK2 lengths in xrRNA1s are longer than that in xrRNA2s for ZIKV, WNV, and other viruses. So it's difficult to draw a general conclusion that the increased Xrn1 resistance would result in more severe flavivirus diseases, enhanced viral fitness and pathogenicity. The significance of the duplication pattern of xrRNAs in the particular virus requires further investigation.

Response Fig. 1. The PK2 length of xrRNA1 are plotted against the PK2 length of xrRNA2 for the various flaviviruses.

10. In the results section authors noted that “.....Although Hstate can directly transit to Lstate under almost all conditions, its reverse reaction, direct transition from Lstate to Hstate, is extremely slow and rarely seen, even at high Mg^{2+} when Hstate is the most stable state. These phenomena strongly support that the folding of xrRNAs follows a definite sequential pathway corresponding to transitions from Lstate to Istate, then to Hstate.....” –

Is this totally correct?

- For example, in WNV-xrRNA1, Fig. 3c, it seems that the folded to unfolded transition always happens through the partially folded state. This should be noted in the Results section with a possible explanation.

- From the traces shown in Fig. 3a-c, the sequential pathway from L to I to H seems not always followed. This part of the results section needs major correction to clarify the folding mechanism in more detailed manner. Again, TDP/TODP plots would greatly help lend statistical significance to the stated observations. In addition, one can estimate (from the rate constants) how likely it is for I to be skipped and a direct L-to-H transition to instead be observed – is this expectation met?

Response #13: Thanks for the reviewer’s suggestions. As suggested, we performed transition density plot (TDP) analysis on the smFRET traces for DENV-xrRNA1, ZIKV-xrRNA1 and WNV-xrRNA1. The results were included in **Figure 3g-i** and Supplementary **Fig. 7d-f**. As Mg^{2+} increases, the number of molecules which transit

between different FRET states decrease, implying that xrRNAs become more stable. According to the transition density plots, xrRNAs can directly transit between low FRET and intermediate FRET, and transit between intermediate FRET and high FRET. But it is rarely seen for xrRNAs to transit from low FRET to high FRET by skipping the intermediate state, implying that folding pathway for flaviviral xrRNAs is sequential. This result is consistent with recent mechanical study on ZIKV xrRNA1 using optical tweezers, which only ~4% molecules form PK2 directly with lower mechanical stability (PMID: 34253909). As for the smFRET traces in **Figure 3a-c**, we have reselected the traces which should be more representative in the revised manuscript.

11. In the Discussion, there are instances where the same result is discussed multiple times. Instead, a considerably more thorough contextualization with respect to the known data on xrRNA should be presented.

Response #14: Thanks for the reviewer's comments. We have revised the main text accordingly.

12. Figure 1 should indicate where the mutations were later made to guide the reader.

Response #15: Many thanks for the reviewer's suggestions. We have modified **Figure 1a** to indicate the mutations and included more details in the Figure legend.

13. Figure 2 and 3 could be combined because they refer to the same experiment.

Response #16: To highlight the significance of UBP-based site-specific fluorescence labeling strategy for large RNAs, we decide not to combine Figure 2 and Figure 3. Furthermore, we add a schematic (new **Figure 2c**) to explain the procedure in more details.

14. Figures 3 and 6: Often with smFRET trace fitting, a hidden Markov model is

overlaid on top of the traces to demonstrate how well the models describe the data. Figures 3A and 6A would be improved with the incorporation of fits displayed over the traces (include example) as that would make the assignments of the three purported states much clearer.

Response #17: As suggested by the reviewer, we have added the HMM-idealized traces over the experimental trace in **Figure 3a-c** and **Figure 6a-c**.

15. Did the authors consider any “static” traces in the derivation of their rate constants? If molecules do not change state over the relatively short observation windows used here, then they might transition during the unobserved next instance, which would skew the reported rate constants.

Response #18: We thank the reviewer for pointing out this likelihood. HMM-based method was used to perform three-state fitting analysis on smFRET data of flaviviral xrRNAs in different Mg^{2+} concentrations. The proportion of molecules undergoing transitions between different states was more than 90% (summarized in **Table S4**). In addition, the average bleaching time for single RNA molecule is ~2.7-4 s, which is usually 2.5-fold longer than the dwell time for xrRNAs in different FRET. Therefore, the contribution of photobleaching to events used in dwell time distributions is < 10% ($e^{-2.5}$), which is acceptable in the smFRET experiments. In summary, the transition rates reported here are reliable.

In addition, can the authors’ quantification of the rate and equilibrium constants be related back to the kinetics and thermodynamics of Xrn1’s degradation action? Xrn1 is a motor protein that converts phosphodiester hydrolysis into a force for forward movement, counteracted by the force resistance of the pseudoknot; alternatively, degradation could be kinetically controlled by the relative rates of opening/closing of the pseudoknot and that of Xrn1’s advance on the RNA. The authors should have this information, and by such an additional, more sophisticated analysis could substantially

strengthen the biological significance and impact of their work.

Response #19: We thank the reviewer for the insightful comments. To relate the kinetics and thermodynamics of Xrn1's degradation action, we performed correlation analysis between the Xrn1 decay rates and free energy in high FRET state or transition rates. While the free energies in high FRET states show positive correlation with decay rates (**Figure 4k**), which the Pearson's correlation constant is 0.8, there is no correlation between the free energies of the intermediate FRET state and the decay rates (data not shown). Beyond that, the Xrn1 decay rates show positive correlation with the transition rates of k_{H-I} and k_{I-L} , but no correlation with that of k_{I-H} and k_{L-I} , indicating that the faster xrRNA escapes from the High FRET state, the easier it is degraded by Xrn1. These results revealed that the degradation of xrRNAs by Xrn1 could be kinetically controlled by the unfolding rates of xrRNAs (**Figure 4g-j**). We added new figures and have discussed the results in the revised manuscript.

16. Their transition state analysis tells the authors that the transition state is more similar to the I state than the H state – they should be more explicit about broader conclusions such as this from their transition state theory in Figures 4 and 6.

Response #20: From **Figure 4** and Supplementary **Fig. 9**, the transition states, the I_{state} and H_{state} are all affected by Mg^{2+} , we couldn't draw the conclusion that the transition states are more similar to the I_{state} than the H_{state} .

17. For Figure 6, the authors should discuss what the significance is that the I state shifts, in some cases quite dramatically, as is the case for the PK2-L3 mutant of WNV.

Response #21: Thanks for the reviewer's suggestions. Tertiary contacts in different regions are usually coupled to guide the cooperative folding of RNAs. As exemplified by previous study on group I ribozyme (PMID: 22500801), it's likely that a single mutation at certain key tertiary interaction motif may disrupt the coupling among different tertiary interactions, hence result in the formation of non-native intermediates which are stabilized by additional base-pairing and base-stacking interactions to

compensate lost tertiary contacts. In **Figure 6**, we mainly discuss the role of tertiary interactions including PK1, PK2 and J2/3 in maintaining the folding and structural integrity of xrRNAs. As reviewer mentioned, the intermediate FRET state (I_{state}) of PK2-L3 mutants shifts significantly compared to WT. Our work implies a delicate balance between Mg^{2+} and PK2 length (strength of a key tertiary motif) in modulating the cooperativity of tertiary interaction motifs to guide the folding and function of xrRNAs. It's speculated that the tertiary interaction mutants such as WNV-PK2-L3 form non-native intermediate structures that are different from WT, which eventually results in the misfolded conformation and can't resist the degradation by Xrn1 even at high Mg^{2+} (**Figure 5g**). We have clarified this point in the revised manuscript.

Minor Concerns:

1. *There are many grammatical issues throughout the manuscript, which at times distract from the message, especially in the Introduction and Discussion sections. These issues must be addressed prior to resubmission. Some examples are listed below:*

a. *Line 64: "Despite of the divergence..." should be "Despite the divergence..."*

b. *Line 103: "...and mutations that disrupt key tertiary interactions including PK1, PK2 and J2/3 using single-molecule Fluorescence Resonance Energy Transfer (smFRET)." is confusing.*

c. *Line 144: "...fluorescence-based Xrn1 decay assay (Supplementary Fig. 3a) was performed..." should be "...a fluorescence-based Xrn1 decay assay (Supplementary Fig. 3a) was performed..." or "...fluorescence-based Xrn1 decay assays (Supplementary Fig. 3a) were performed..."*

d. *Line 177: "intent" should be "intend".*

e. *Line 440: "An ensemble description of RNA structure has been increasingly applied to clarify RNA folding, conformational dynamics and understand RNA function."*

f. *Line 446: "Recent biophysical studies showed that ZIKV-xrRNA1 form alternative" should be "forms alternative".*

g. *These are among many other examples throughout the manuscript.*

Response #22: We thank the reviewer for the comments and suggestions and we have carefully checked the manuscript, fixed the grammatical issues in the revised text.

2. *On page 4, lines 73 and 75, it is unclear what the authors mean by “mechanical anisotropy”.*

Response #23: We are sorry for this confusion. In the revised text, we explain “mechanical anisotropy” at the first place as “which responds to mechanical stretching in a direction-dependent manner”. The mechanical anisotropy of ZIKV xrRNA1 has been predicted previously and confirmed recently by us (PMID: 33127896) using single molecule nanopore technique.

3. *On page 5, line 91, the authors refer to the “undocked” and “docked” conformations. It is unclear at this point what that means. Could the authors expand on this statement, and point more clearly in Figure 1b as to what they mean by “undocked” and “docked” conformations?*

Response #24: We feel sorry about the confusion and have revised the main text and **Figure 1b** accordingly. The key tertiary interaction motif, PK2 (pseudoknot 2) in the crystal structure of ZIKV-xrRNA1 (PDB: 5TPY), is formed between L3 and S4 (orange region in **Figure 1b**), but not formed in the crystal structure of MVE-xrRNA2 (PDB: 4QPV). Based on the distance measurements in **Figure 1b**, ZIKV-xrRNA1 and MVE-xrRNA2 are assumed to adopt fully folded (PK2 is formed) and partially folded (PK2 is not formed) conformations, respectively. We don't use the terms of “docked” and “undocked” conformations in the revised manuscript.

4. *For Figure 1f, the authors do not clearly state what strategy (either the gel assays or fluorescence assays) is used to produce the decay rate curves. This should be made clear in both the figure legend and the text. In addition, the positions of panels c and f should be switched for a more canonical layout.*

Response #25: We thank the reviewer's comments and suggestions. As suggested, we have revised the text and figure legend regarding to **Figure 1f**, switched panels c and f accordingly.

5. *In Supplementary Figure 3, it is unclear what the decay products are in the denaturing PAGE gels. Are these the end products for each [Mg²⁺] at 60 minutes? This should be noted in the legend.*

Response #26: As suggested, we have clarified the decay products in the figure legends of Supplementary **Figures 4, 5, 6** and **8** in the revised version. The denaturing PAGE gels show the end products of the decay reactions in different Mg²⁺ concentrations at 60 min.

6. *In Figure 4, it might be helpful to the audience if the PK2 length was mentioned in the legend or titles of the plots, to help remind readers that these xrRNAs represent "short", "medium", and "long" PK2s.*

Response #27: As suggested, we have added PK2 lengths of the respective xrRNAs to the title of all appropriate Figures or Supplementary Figures in the revised version.

7. *The colors used in Figure 4c-f for 1 mM and 100 mM Mg²⁺ are very similar, and hard to distinguish. The authors may want to consider adopting the color scheme used in Figure S7, which provides much more distinction.*

Response #28: We thank the reviewer for pointing out this. We have changed the color schemes in **Figure 4c-f** for better distinction.

8. *The normalization strategy (i.e., the equation used) for the Xrn1 decay kinetics assay should be provided or at least described in words.*

Response #29: As suggested, we have described the normalization strategy in the **METHODS** section as "The fluorescence were normalized to controls without Xrn1

treatment based on the equation: $F = (F_{\text{exp},t}/F_{\text{exp},t=0})/(F_{\text{con},t}/F_{\text{con},t=0})$. F_{exp} and F_{con} refer to the fluorescence of the experimental group (+Xrn1) and control group in the absence of Xrn1 (-Xrn1), respectively.”

9. In the Methods section, it looks like “°C” is not recognized as a character for the PDF version of the manuscript. It appears as a square. This should be amended in future versions.

Response #30: We have amended the symbol of “°C” in the revised version.

10. The $k_B T$ values should (also) be given as more canonical kcal/mol values.

Response #31: As suggested, we have converted the $k_B T$ values to kcal/mol values in **Figure 4d-f** and Supplementary **Figure 9d-f**.

Response to reviewer 2

Referee #2:

Exonuclease-resistant RNA pseudoknots (xrRNAs) have attracted considerable interest in recent years because of their importance in flaviviruses where they first discovered, and their usefulness in RNA technologies. This manuscript very thoroughly describes how the length of pseudoknot 2 (PK2) affects the stability of xrRNA tertiary structure and the ability to resist degradation by Xrn1 exonuclease. This follows up the NComm 2020 paper by the same authors on the mechanical stability of xrRNA (using a nanopore).

The strength of this study is its thoroughness – the authors begin by comparing 11 (!) xrRNA motifs from different viruses, and then investigate three examples in more depth. These examples have different numbers of base pairs in PK2. It is commendable that the authors looked at so many different natural xrRNA sequences in order to accurately assess the importance of PK2 length. Another strength of this study is that the authors studied the folding of the xrRNAs with exonuclease assays and with biophysical

methods, SAXS and smFRET, that provide complementary information about the folding equilibrium and kinetics. Finally, they use targeted mutations to evaluate the cooperativity between PK2 and other elements of xrRNA tertiary structure.

The final conclusion is not surprising – a longer PK2 pairing correlates with a more stable fold and better exonuclease resistance. That PK2 is important for exonuclease resistance was shown in the earliest papers on this motif and the authors showed in their 2020 paper that PK2 is particularly important for mechanical stability. The authors don't really connect their findings to the biological roles of these motifs in different viruses, although presumably other authors could do that. The main value of this study may be for xrRNA engineering. Perhaps the authors could comment on that a bit more.

Although a few points listed below require clarification, the experiments were well-done, and the manuscript is written clearly.

Response #32: We thank the reviewer very much for the nice comments and constructive suggestions on our work. We have addressed the concerns point-by-point as below.

Specific comments:

1. The SAXS experiments nicely show that all of the natural xrRNAs fold to a compact structure in 5 mM MgCl₂. The authors emphasize that the ones with the longest PK2 are also able to fold in EDTA, based on R_g and D_{max}. However, the reduced Kratky plots in Fig. 5c show that even the most stable sequence is very clearly less compact in EDTA than it is in 5 mM MgCl₂. This is presumably due to transient unfolding of the RNA tertiary structure but could also arise from non-native structure as the authors allude to later. I think the authors should revise their text on lines 139-141 to acknowledge this possibility. It is possible that none of the xrRNAs achieve the true native structure without some Mg²⁺, although they can get close if PK2 is stable enough. Evidence for this is also in Fig. 5ab.

Response #33: We agree with the reviewer that Mg²⁺ is required to achieve stable

native folding for xrRNAs, including that with long PK2. Though the R_g and D_{max} of xrRNAs with long PK2 in EDTA can get close to that in 5 mM Mg^{2+} , they were still less compact and more dynamic in conformational ensemble. This was also supported by smFRET data of WNV-xrRNA1 in low Mg^{2+} such as 0.001 mM. As suggested, we have revised the text accordingly.

2. The authors use a non-standard base to introduce the Cy3 dye next to PK2 for smFRET studies. They state that this modification has only a minor effect on the stability of the RNA tertiary structure, but the data showing this in Fig. S4 are very brief. What length of Xrn1 treatment and Mg^{2+} concentration was used to test the stabilities of the modified RNAs? How sensitive was this test, and is it a fair comparison without a leader that can be grasped by Xrn1?

Response #34: To investigate the effects of dye labeling on xrRNAs in more details, in addition to the denaturing PAGE gel analysis, we also performed the more sensitive fluorescence-based Xrn1 decay kinetics assay for Cy3-labeled DENV2-, ZIKV- and WNV- xrRNA1s in the presence of 1, 5 and 10 mM Mg^{2+} , respectively. These new results are shown in Supplementary **Fig. 6d-f**. Each of the Cy3-labeled xrRNA1s used here possesses a 6-nt leader sequence (4A+2G) at the 5' end which is long enough to be grasped by Xrn1 (PMID: 21362555). Both the fluorescence-based assay and the denaturing PAGE analysis of the end-product of Cy3-labeled xrRNAs show that fluorescent labeling causes minimal effect in 5 or 10 mM Mg^{2+} , but some perturbations in 1 mM Mg^{2+} on xrRNAs' activity.

3. The potential for interference by the fluorophore is relevant to interpreting the smFRET experiments. In general, these data are very lovely, but as the authors note on lines 235-236 and 258, even the most stable xrRNA studied continues to fluctuate between the folded and unfolded states in 100 mM $MgCl_2$. Conditions that yield maximum Xrn1 resistance correlate with as little as 45% occupancy of the folded state. These observations beg the question of how fluctuating xrRNAs can resist the

exonuclease – is there some minimum dwell time in the unfolded state required to let Xrn1 by? Or, is the fluorophore is making the structures more dynamic, leading to this apparent discrepancy? If the latter were true, I don't think it would undermine the point of this paper (a comparison of PK2 lengths), but it would be important to measure any perturbation to arrive at the proper interpretation. (NB: it is possible for the fluorophore to perturb the folding and unfolding rate constants without a large change in the folding equilibrium.)

Response #35: Many thanks for the reviewer's insightful comments and suggestions. The fluorescence-based Xrn1 decay kinetics assay suggested that the fluorophore labeling causes some minor effects on the activity of xrRNAs, especially at lower Mg^{2+} . Currently, it's difficult to find a fluorophore or a labeling method that will not cause any perturbations to the system when external fluorophores are incorporated into the RNA. To understand the relevance of the results from different analysis, we performed correlation analysis of the free energy in high FRET states or transition rates between different states derived from smFRET experiments with the decay rates in bulk Xrn1 decay kinetics assay. As shown in **Figures 4g-j**, the Xrn1 decay rates are positively correlated with the transition rates of k_{H-I} and k_{I-L} , but not for k_{L-I} and k_{I-H} , indicating that the faster xrRNA escapes from the H_{state} or I_{state} , the easier it is degraded by Xrn1. In addition, the free energies of H_{state} for xrRNAs exhibit significant positive correlation with decay rates (**Figure 4k**). These analyses reveal that the degradation of xrRNAs by Xrn1 is kinetically controlled by the transitions of xrRNAs from H_{state} to I_{state} or L_{state} . In other words, Xrn1 decay of xrRNA is regulated by the unfolding rate of the ring-like structure.

4. The FRET efficiencies are fit to three states in Fig. 3, but the data suggest there may exist one or more "hidden" intermediate states that the authors should consider (or at least discuss). First, the wt xrRNAs seem to form folded (high FRET) states with different lifetimes. Sometimes this shows up as fluctuations in the Cy5 channel, and sometimes as FRET changes that are short lived (eg ZIKV in 1 mM MgCl₂). Second,

the average FRET values of the L, I and H populations shift with MgCl₂ concentration in the histograms (Fig. 3d-f), which should not be the case if the system is merely sampling three states. This possibility becomes more apparent for the mutants in Fig. 6. I recommend that the authors look more carefully at the data and consider whether a more complicated folding model would help them better explain the effects of the mutations and the link between folding dynamics and exonuclease resistance.

Response #36: Many thanks for the reviewer's suggestions. We agree with the reviewer that the average FRET values of the L, I and H populations shift with Mg²⁺ concentrations in the histograms. This may be due to that Mg²⁺ binding to the RNA promotes a global compaction of the xrRNA fold on a time-averaged basis, similar phenomenon has been observed on the twister ribozyme (PMID: 28598157).

The distribution of dwell time in a particular FRET state usually follows exponential decay curve, which leads to the existence of long and short FRET events.

We have looked carefully at the smFRET data and consider that the three-state folding model for xrRNAs is more appropriate for the following reasons. Kieft et al proposed a three-state folding model for xrRNAs on the basis of the crystal structure of MVE xrRNA2, in which the PK2 was not formed yet the ring largely formed. The model indicated that PK1 and junction folds first to form the ring around the 5' end, this also positions the L3 and S4 bases to form PK2 (PMID: 26399159). This model was also supported by a recent study on ZIKV xrRNA1 by using optical tweezers (PMID: 34253909), in which the secondary structure folds first, then the 5' end interacts with the nucleotides in 3WJ to form the threaded intermediates, finally the L3-S4 PK2 formed to close the ring. In consideration of the previous proposed three-state folding pathway for xrRNAs and the minimal state principle when using HMM-based fitting analysis, we choose three-state model to elucidate the mechanisms underlying the folding dynamics and Xrn1 resistance.

5. In Figure 6g, the authors state that the mutant xrRNAs are misfolding, yet this is not

shown in the free energy landscape. This diagram should be modified to indicate a fourth conformational state in keeping with their interpretation.

Response #37: Thanks for the reviewer's suggestion, we have modified the free energy diagram and added a fourth conformational state in **Figure 7** (previous **Figure 6g**).

6. What transmission frequency (barrier free rate constant) was assumed for the folding reactions when building the free energy landscapes (line 642), and how was this chosen?

Response #38: The actual energy barriers between states cannot be determined via our experiments. In our energy landscapes, the energy barrier from state a to b was shown as $-k_B T \cdot \ln(k_{a \rightarrow b}) + 1.8k_B T$. The arbitrary value of $1.8k_B T$ was used to have higher energy barriers than free energies of three FRET states and to make energy landscapes more suitable for visualization. We have described the details in the **Methods** section of the revised manuscript.

7. The authors should provide more information about the numbers of molecules used for each smFRET experiment and the numbers of observations, as well as the error analysis. Perhaps the number of molecules can be included in the supplemental tables.

Response #39: Thanks for reviewer's suggestion. As suggested, the number of molecules for each smFRET experiment has been added in the relevant Figures. The error analysis from three independent experiments has also been included in the revised manuscript.

Response to reviewer 3

Referee #3:

The manuscript of Xiaolin Niu and co-authors reports the study on thermodynamic stability and exoribonuclease resistance of flaviviral xrRNAs, the conserved XRN-1-resistant RNA elements in 3'UTRs of flaviviruses responsible for biogenesis of viral noncoding RNA (sfRNA). The exoribonuclease resistance of xrRNAs is achieved due to

the unique ring-like structure of these elements, which is determined by canonical and noncanonical base pairing interactions within the structure. The most critical among them are two pseudoknots (small pseudoknot PK1 and typically large pseudoknot PK2) and interactions of unpaired C nucleotide (J2/3). By using the range of biochemical and biophysical techniques the authors demonstrated that formation of PKs are Mg^{2+} -dependent, however the concentration of magnesium required for PK formation depends on length of PK2 with longer PKs requiring less Mg^{2+} . Authors also demonstrated that xrRNAs exist in three conformational states – the stable XRN1-resistant fold and two transitional folds that are not XRN1 resistant. xrRNAs spend half of their time in XRN1-resistant state and present only transiently in nonrestraint states. Authors demonstrated that higher Mg^{2+} concentration and longer PK2 stabilise XRN1 resistant state. As the result xrRNAs with longer PK2 (7nt) can tolerate mutations in PK1 and J2/3. Although the requirement of Mg^{2+} for XRN1 resistance of xrRNAs is known (PMID: 24744377), the rest of the results represent novel findings that improve our mechanistic understanding of XRN-1 resistance.

Response #40: We thank the reviewer for the positive comments on our manuscript and address the concerns point-by-point as below.

However, the biological implications of these purely in vitro findings are not investigated or even discussed in the manuscript. These include:

1. What are the implications of Mg^{2+} concentration on different conformational states of xrRNAs in the physiologically relevant Mg^{2+} concentrations in virus-infected cells (0.5-1mM, e.g. PMID: 29920012)

Response #41: Mg^{2+} has been known to be essential for the structure and folding of RNAs. In our manuscript, we performed smFRET experiments for flaviviral xrRNAs with different PK2 lengths in various Mg^{2+} concentrations ranging from 0.001 to 100 mM as previous studies (PMID: 28825710, 28920931). In this way, we have demonstrated how Mg^{2+} and PK2 length together sculpture the dynamic conformational

ensembles of xrRNAs in dilute solutions. These *in vitro* results may not apply directly to the cellular conditions in virus-infected cells, which are much crowded and the physiologically relevant Mg^{2+} concentrations are about 0.5 – 1 mM, however, they still shed insights into the physiological roles of Mg^{2+} to xrRNAs' functions. For example, the smFRET data indicated that xrRNAs with short PK2 tend to transit from H_{state} to I_{state} or L_{state} , while xrRNAs with long PK2 are more stable in H_{state} at low Mg^{2+} concentrations, as the stable H_{state} confers Xrn1 resistance, it's expected that xrRNAs with long PK2 are responsible to produce more sfRNAs in virus-infected cells.

2. What are the roles of different conformational states of xrRNA in virus replication and virus-host interactions?

Response #42: The smFRET analysis can be used to characterize dynamic conformational ensemble of RNA, which reveals three distinct FRET states (L_{state} , I_{state} and H_{state}), corresponding to the unfolded, partially folded and folded conformations of the flaviviral xrRNAs. The folded conformation of xrRNAs is capable of stalling Xrn1, resulting in the formation and accumulation of sfRNAs in the infected cells. sfRNAs have been implicated to impact many cellular processes including flavivirus replication, cytopathicity and pathogenicity. The transitions of folded to partially folded, or unfolded conformations, in other words, the conformational dynamics are important to the function of xrRNAs. For example, xrRNA degradation by Xrn1 requires the escape from folded conformation to partially folded or unfolded conformations (our updated **Figs. 4g-j**). Also, during the replication or translation processes catalyzed by RNA-dependent RNA polymerase or ribosome, the xrRNA need to undergo mechanical unfolding.

3. Is there any correlation between PK2 length and pathogenicity or epidemiological fitness of the viruses?

Response #43: The correlation between PK2 length and the pathogenicity or

epidemiological fitness of flaviviruses is an important, interesting but challenging question to be answered with further studies. Such studies require intensive efforts, thus is beyond the focus of our current manuscript. However, our findings can provide insights into the question, which is discussed in the revised manuscript (see also **Response #12**).

An intriguing feature in most mosquito-borne flavivirus (MBFV) is the presence of duplicated xrRNAs (xrRNA1 and xrRNA2) in the 3' UTR of viral genomes. This duplication pattern has been linked to the generation of multiple sfRNAs in flaviviruses, adaptation to different hosts and efficient transmission, etc. We analyze the PK2 lengths of xrRNA1 and xrRNA2 across different flaviviruses and find no clear pattern. For example, in DENV1 and DENV2, the PK2 lengths in xrRNA1s are shorter than that in xrRNA2s, however, the PK2 lengths in xrRNA1s are longer than that in xrRNA2s for ZIKV, WNV, and other viruses. So it's difficult if not impossible to draw a general conclusion about the correlation between PK2 length and the pathogenicity or epidemiological fitness of flaviviruses.

4. Is there an evolutionary trend in increasing PK2 length?

Response #44: Whether there is an evolutionary trend in increasing PK2 length is an interesting question, which needs further study.

The existence of duplicated xrRNA structures (xrRNA1 and xrRNA2) at near the beginning of 3'UTR of genomic RNA is a common feature for all known flaviviruses, which has been linked to host adaptation. Although the two xrRNAs share conserved primary sequences and secondary structures, the PK2 lengths in xrRNA1 and xrRNA2 are not identical in a particular flavivirus. Our statistics show that there is no clear pattern for the PK2 length in xrRNA1 and xrRNA2 across different viruses (see also **Response #12**). There may be certain restraints for the PK2 length in a particular virus. Recent studies on ZIKV and DENV2 revealed that both xrRNAs in 3'UTR are enhancers for viral replication in human cells while play opposite roles in the mosquito

host. The mutation or depletion of xrRNAs with longer PK2 such as ZIKV xrRNA1 (4 bp) or DENV2 xrRNA2 (4 bp) impairs the viral replication in human cells, whereas enhances the viral replication in mosquito cells and increase the fitness and adaptation of virus to mosquito hosts. Additionally, previous studies found that when DENV2 virus transport from human to mosquito hosts, mutations accumulate in xrRNA2 (4 bp) with longer PK2 to attenuate the amount of sfRNA produced (PMID: 32581095, 25635835). In combination with our results that xrRNAs with short PK2 are less efficient in stalling Xrn1 than xrRNAs with long PK2, it's reasonable to speculate that xrRNAs with shorter PK2 in 3'UTR are required to produce moderate amount of sfRNAs in mosquito host to achieve a balance between the mosquito survival and efficient replication of flavivirus for successful onward transmission. These hypothesizes require further investigation in future study.

REVIEWERS' COMMENTS

Reviewer #1 (Remarks to the Author):

The authors have done an admirable job with their extensive revision. Their data are now even more comprehensive, since important controls were added, and represent a significant advance in the field. All reviewer suggestions were satisfactorily accommodated, strengthening the overall story significantly. The authors present their data with well-organized figures and descriptions.

Reviewer #2 (Remarks to the Author):

The authors have addressed the comments of the previous reviewers very well and the revised manuscript is much improved over the original version. The main improvements are clearer and more quantitative data analysis, including transition density plots, and additional experiments on ZIKV and WNV xrRNAs containing mutations in pseudoknot 2. The results of these new experiments support the conclusion that unfolding of pseudoknot 2 and the surrounding tertiary structure is what allows the exonuclease to degrade the RNA. If this structure is stable, then the xrRNA is better at resisting degradation. The authors now provide plots showing how each step of folding or unfolding correlates with the rate of decay by Xrn1 in Fig. 4. Overall, this paper contains a very large amount of data, and provides a comprehensive picture of the in vitro folding kinetics of these xrRNAs. This study certainly advances our understanding of this class of non-coding elements on viral RNA genomes.

The authors were not able to fully address the following points raised in the previous review:

- (1) All three reviewers noted the weak connection between the in vitro data and biological function. The authors discuss the importance of these pseudoknotted structures to flavivirus fitness, but the data in the revised paper still focus entirely on the folding pathway and stability of the xrRNA domain in vitro. I think this is OK but it is a limitation.
- (2) The additional PK2 mutants were very helpful for establishing the importance of PK2 length (Fig. S5), but these data are not incorporated into any of the main figures, so their impact is lost. I think the authors should either make Fig. S5 a main figure, or at least incorporate the results into appropriate summary figures, such as the bar graphs in Fig. 1d,e. Or, they could introduce a separate bar graph comparing the mutants from Fig. S5, in lieu of Fig. 1e. D max and Rg correlate with each other, so showing both parameters in Fig. 1 doesn't really add anything.
- (3) The authors attempt to correlate folding stability and Xrn1 decay (Fig. 3m, Fig. 4g, 4h, 4k), but the correlation is poor although the trendline is in the right direction. This poor correlation raises the possibility that additional factors matter. Would they have gotten a better correlation if they could use the human Xrn1? I don't think that they should repeat the decay experiments, but the fact that unfolding rate or stability doesn't fully explain Xrn1 decay is something worth noting in their discussion.
- (4) The authors show that the fluorophores only minimally perturb the resistance to Xrn1 at high Mg²⁺ (Fig. S6). This is helpful, but not the most sensitive test (the Mg²⁺ midpoint would be more sensitive), so the possibility remains that the poor correlation noted above arises from the perturbations to the RNA folding dynamics by the fluorophores.

On balance, this very thorough study of the folding of xrRNAs sheds lights on their function in flavivirus processing. The points above are minor issues, not major problems, in my view. However, I note them in the event that the authors are asked to revised their manuscript once more.

Reviewer #3 (Remarks to the Author):

The authors have addressed all my concerns and improved the manuscript

Response to reviewer 2

Referee #2:

The authors have addressed the comments of the previous reviewers very well and the revised manuscript is much improved over the original version. The main improvements are clearer and more quantitative data analysis, including transition density plots, and additional experiments on ZIKV and WNV xrRNAs containing mutations in pseudoknot 2. The results of these new experiments support the conclusion that unfolding of pseudoknot 2 and the surrounding tertiary structure is what allows the exonuclease to degrade the RNA. If this structure is stable, then the xrRNA is better at resisting degradation. The authors now provide plots showing how each step of folding or unfolding correlates with the rate of decay by Xrn1 in Fig. 4. Overall, this paper contains a very large amount of data, and provides a comprehensive picture of the in vitro folding kinetics of these xrRNAs. This study certainly advances our understanding of this class of non-coding elements on viral RNA genomes.

Response #1: We appreciate the reviewer's positive comments and constructive suggestions on our work and have addressed all the minor concerns point-by-point as below.

The authors were not able to fully address the following points raised in the previous review:

(1) All three reviewers noted the weak connection between the in vitro data and biological function. The authors discuss the importance of these pseudoknotted structures to flavivirus fitness, but the data in the revised paper still focus entirely on the folding pathway and stability of the xrRNA domain in vitro. I think this is OK but it is a limitation.

Response #2: Thanks for reviewer's suggestion. The most important biological functions of xrRNA relates to its *in vitro* Xrn1 (or other exoribonucleases) resistance and cellular sfRNA (subgenomic flaviviral RNA) formation. Our study reveals an

important regulatory role of PK2 length in modulating the structure, conformational dynamics and Xrn1 resistance of flaviviral xrRNAs. Though conducted *in vitro*, these results could provide mechanistic insights into the *in vivo* functions of xrRNAs, which require further investigation in future studies.

(2) The additional PK2 mutants were very helpful for establishing the importance of PK2 length (Fig. S5), but these data are not incorporated into any of the main figures, so their impact is lost. I think the authors should either make Fig. S5 a main figure, or at least incorporate the results into appropriate summary figures, such as the bar graphs in Fig. 1d,e. Or, they could introduce a separate bar graph comparing the mutants from Fig. S5, in lieu of Fig. 1e. D max and Rg correlate with each other, so showing both parameters in Fig. 1 doesn't really add anything.

Response #3: We thank the reviewer's positive comments and constructive suggestions and have reorganized the **Fig.1** and **Fig. S5**.

(3) The authors attempt to correlate folding stability and Xrn1 decay (Fig. 3m, Fig. 4g, 4h, 4k), but the correlation is poor although the trendline is in the right direction. This poor correlation raises the possibility that additional factors matter. Would they have gotten a better correlation if they could use the human Xrn1? I don't think that they should repeat the decay experiments, but the fact that unfolding rate or stability doesn't fully explain Xrn1 decay is something worth noting in their discussion.

Response #4: Thanks for the reviewer's comments and suggestions. The Xrn1 decay rates of flaviviral xrRNAs show positive correlation with transition rates k_{H-I} and k_{I-L} derived from smFRET experiments, though the correlation is not perfect. There could be several reasons. Firstly, Xrn1 decay kinetics assays were conducted at 37 °C while smFRET experiments were performed at 25 °C, the temperature difference between two experiments may account for the poor correlation. Secondly, the fluorophores may indeed cause minor perturbations on the folding and dynamics of xrRNAs, which can't be easily distinguished in high Mg^{2+} (**Fig. S6**). Thirdly, as reviewer mentioned, the Xrn1 exonuclease used in our decay kinetics assay derives from yeast but not

human, which may also have a minor effect. We have revised the main-text and discussed the possible reasons for the poor correlation.

(4) The authors show that the fluorophores only minimally perturb the resistance to Xrn1 at high Mg²⁺ (Fig. S6). This is helpful, but not the most sensitive test (the Mg²⁺ midpoint would be more sensitive), so the possibility remains that the poor correlation noted above arises from the perturbations to the RNA folding dynamics by the fluorophores.

Response #5: Thanks for reviewer's comments. Our Xrn1 decay kinetics assay on Cy3-labeled xrRNAs shows that the fluorophores have a minor effect on the activity of xrRNA1 in high Mg²⁺ (5 or 10 mM), but cause some perturbations in 1 mM Mg²⁺ (**Fig. S6**). We agree with the reviewer that the fluorophores may indeed have some effects on the xrRNA's folding dynamics, which can't be easily distinguished in high Mg²⁺ and finally lead to the poor correlation between transition rates and decay rates. We have revised the main-text.

On balance, this very thorough study of the folding of xrRNAs sheds lights on their function in flavivirus processing. The points above are minor issues, not major problems, in my view. However, I note them in the event that the authors are asked to revised their manuscript once more.